# Selection of genotypes harbouring mutations in the *cytochrome b* gene of *Theileria annulata* is associated with resistance to buparvaquone

**Selin Hacılarlıoglu**[1◉], **Huseyin Bilgin Bilgic**[1‡], **Serkan Bakırcı**[1‡], **Andrew Tait**[2‡], **William Weir**[2‡], **Brian Shiels**[2‡], **Tulin Karagenc**[1◉]*

1 Faculty of Veterinary Medicine, Department of Parasitology, Aydın Adnan Menderes University, Isıklı, Aydın, Türkiye, 2 School of Biodiversity, One Health and Veterinary Medicine, College of Medicine, Veterinary and Life Sciences, University of Glasgow, Glasgow, United Kingdom

◉ These authors contributed equally to this work.
‡ These authors also contributed equally to this work.
* tkaragenc@adu.edu.tr

**Data Availability Statement:** All relevant data are within the paper and its Supporting Information files.

## Abstract

Buparvaquone remains the only effective therapeutic agent for the treatment of tropical thei-leriosis caused by *Theileria annulata*. However, an increase in the rate of buparvaquone treatment failures has been observed in recent years, raising the possibility that resistance to this drug is associated with the selection of *T. annulata* genotypes bearing mutation(s) in the *cytochrome b* gene (*Cyto b*). The aim of the present study was: (1) to demonstrate whether there is an association between mutations in the *T. annulata Cyto b* gene and selection of parasite-infected cells resistant to buparvaquone and (2) to determine the frequency of these mutations in parasites derived from infected cattle in the Aydın region of Türkiye. Susceptibility to buparvaquone was assessed by comparing the proliferative index of schizont-infected cells obtained from cattle with theileriosis before and/or after treatment with various doses of buparvaquone, using the 3-(4,5-dimethyl thiazol-2-yl)-2,5-diphenyl tetrazolium bromide (MTT) colourimetric assay. The DNA sequence of the parasite *Cyto b* gene from cell lines identified as resistant or susceptible was determined. A total of six non-synonymous and six synonymous mutations were identified. Two of the nonsynonymous mutations resulted in the substitutions V135A and P253S which are located at the putative buparvaquone binding regions of cytochrome b. Allele-specific PCR (AS-PCR) analyses detected the V135A and P253S mutations at a frequency of 3.90% and 3.57% respectively in a regional study population and revealed an increase in the frequency of both mutations over the years. The A53P mutation of *TaPIN1* of *T. annulata*, previously suggested as being involved in buparvaquone resistance, was not detected in any of the clonal cell lines examined in the present study. The observed data strongly suggested that the genetic mutations resulting in V135A and P253S detected at the putative binding sites of buparvaquone in cytochrome b play a significant role in conferring, and promoting selection of, *T. annulata* genotypes resistant to buparvaquone, whereas the role of mutations in *TaPIN1* is more equivocal.

**Funding:** This work was funded by the Wellcome Trust (https://wellcome.org/) awarded to TK and AT (Grant number: 075820/A/04/Z) for an integrated approach for the development of sustainable methods to control topical theileriosis. The funders had no role in study design, data collection and analysis, decision to publish, or preparation of the manuscript.

**Competing interests:** The authors have declared that no competing interests exist.

## Introduction

Tropical theileriosis, caused by the protozoan parasite *T. annulata*, is an economically important bovine disease, transmitted by several species of ixodid ticks of the genus *Hyalomma* [1]. The disease is widespread in North Africa, Southern Europe, India, the Middle East and Asia. Compared to local cattle breeds, exotic breeds are more susceptible to the disease with a mortality rate of 40 to 60% [2].

Current efforts to control tropical theileriosis are based on three main strategies. The first is control of vector ticks, which is expensive and has food and environmental safety problems. The second is immunisation of cattle with infected cell lines attenuated for virulence by long term passage *in vitro* [3–5]. However, the long-term effectiveness of vaccines in endemic regions is questionable and the influence of vaccination on the diversity of field parasite populations remains unknown [6]. The third, most widely applied method is the treatment of acutely infected animals with buparvaquone, which is a hydroxynaphthoquinone antiprotozoal drug. Buparvaquone is the only highly effective drug available against *Theileria* spp. and has been used for the treatment of the disease since the 1980s [7]. However, an increase in the rate of buparvaquone treatment failure has been reported in a number of countries, including Tunisia [8,9], Iran [10], Sudan [11,12] and Egypt [13]. Both the mode of action and the mechanism of resistance to buparvaquone are still not fully known. It was postulated that buparvaquone disrupts the mitochondrial electron transport chain at the cytochrome $bc_1$ complex, as demonstrated for the related drug atovaquone [14] and the 1,4-naphthoquinone family of hydroxynaphthoquinones [15]. Studies addressing resistance to atovaquone in several parasites, including *Plasmodium falciparum* [16,17], *Toxoplasma gondii* [18,19] and *Pneumocystis carinii* [20,21], demonstrated that mutations in the mitochondrial *cytochrome b* gene (*Cyto b*) confer resistance to the drug. Based on the structural similarities between buparvaquone and atovaquone, it was logical to propose that a mutation in *Cyto b* of *T. annulata*, particularly at the binding site of ubiquinone, would be associated with resistance to buparvaquone [8–11].

It is also thought that buparvaquone acts as a peptidyl-prolyl isomerase PIN1 inhibitor. It has been reported that the parasite-encoded peptidyl-prolyl isomerase of *T. annulata* (*TaPIN1*) is secreted into the host cell and modulates oncogenic signalling pathways [22]. The *TaPIN1*-induced transformation process is inhibited by buparvaquone while a mutation (A53P) at residue 53 of the catalytic loop of *TaPIN1* reverses the ability of buparvaquone to inhibit PIN1 activity [22]. In further studies, the A53P mutation was identified in some, but not all, buparvaquone-resistant isolates from both Tunisia and Sudan [12,22]. On the basis of these observations it was proposed that the A53P mutation in *TaPIN1* could be involved in resistance to buparvaquone in field populations of the parasite [12,22], but the strength of this association requires further validation.

The majority of previous studies investigating parasite resistance to anti-theilerial drugs have been conducted using blood samples from animals that are unresponsive to treatment with buparvaquone [10,11,13]. However, it is likely that these samples comprise a mixture of sensitive and resistant parasite populations, thus making it difficult to conclusively identify which parasite genotypes, and mutations in the *Cyto b* gene or *TaPIN1* gene, are most strongly linked to buparvaquone resistance. In the present study, we examined the susceptibility of field-derived isolates to buparvaquone by comparing the proliferative index of *T. annulata* schizont-infected parental and clonal cell lines exposed to different buparvaquone doses using the 3-(4,5-dimethyl thiazol-2-yl)-2,5-diphenyl tetrazolium bromide (MTT) colourimetric assay. Following genotyping of both resistant and susceptible cell lines by available satellite markers, the *T. annulata Cyto b* and *TaPIN1* genes were sequenced to determine whether putative buparvaquone resistance-associated mutations were present. The frequency of the

observed mutations in field parasite populations was then assessed using an allele-specific PCR to investigate the emergence and spread of drug resistance in the field.

## Materials and methods

### Parasite material

A total of 140 *T. annulata* schizont-infected cell lines obtained from animals in the acute phase of theileriosis and 168 *T. annulata* piroplasm-positive blood samples obtained from healthy carrier animals were used in the present study. These samples were gathered between 1998 and 2012 from different farms located within nine different provinces (Center, Söke, Germencik, Kocarlı, İncirliova, Cine, Akçaova, Kösk and Nazilli) of Aydın in Western Türkiye, where tropical theileriosis is endemic.

In order to obtain *T. annulata* schizont infected cell lines, 10 mL blood samples were collected into heparinised tubes from cattle showing clinical signs of the disease before and/or after buparvaquone treatment(s). *In vitro* establishment of schizont-infected cells from peripheral mononuclear cells (PBM) was carried out as previously described [23]. Once the cell line established, a code was given to each isolate according to the site where the blood samples were taken. The letters M, S, G, CN, AC, C, A, K and N stand for the provinces of Center, Söke, Germencik, Kocarlı, İncirliova, Cine, Akçaova, Kösk and Nazilli, respectively. The numbers following these letters indicate the identification number of the animal from which the *T. annulata* schizont cell culture was established, *viz.* A10 stands for the 10[th] animal from the Akçaova province. This was followed by the timing (BT: before or AT: after treatment) and the number of buparvaquone treatment. Accordingly, A10/BT stands for the isolate obtained from the 10[th] animal in Akçaova province before the buparvaquone treatment. Similarly, A21/AT4 stands for the isolate obtained from the 21[st] animal in Akçaova province after the 4[th] buparvaquone treatment.

In an attempt to determine the frequency of resistant parasites in Aydın region, whole blood samples were obtained from a total of more than five hundred, randomly selected healthy cattle in the selected sampling sites. Of these, a total of 168 were found to be positive for *T. annulata* by PCR using *Cyto b* gene primers as described previously [6].

A map illustrating the geographical location of the sampling sites is shown in supporting information (S1 Fig) with isolate details summarised in S1 Table.

### Tetrazolium dye (MTT) colourimetric assay

Susceptibility of *T. annulata* cell line isolates (S1 Table) to buparvaquone was evaluated by using an MTT colourimetric assay to determine the proliferative index of infected culture under various doses of buparvaquone, as described [24] but with slight modification. Briefly, schizont-infected cell lines were cultivated in RPMI 1640 (Gibco, UK) supplemented with 10% Newborn Calf Serum (NCS) (Gibco, UK), 2 mM glutamine, 100 µg/mL streptomycin and 100 IU/mL penicillin. A total of $1 \times 10^5$ viable cells in the active growth phase [23] were dispensed into 48-well plates. Stock buparvaquone (ALKE Healthcare Products Inc., Türkiye) solution (at 4000 ng/mL) was prepared just prior to use, as previously described [25]. Two-fold dilutions of buparvaquone were prepared in complete medium to obtain a range of twelve different drug concentrations. Cells were exposed to buparvaquone in doses ranging from 0.4 to 1,000 ng/mL in triplicate for each dose. Cells cultured in the absence of buparvaquone served as negative control. *Theileria annulata* / Ankara (D7), a clonal cell line known to be susceptible to buparvaquone, was used as a drug-sensitive control parasite population.

All cultivated test and control cell lines were maintained under experimental conditions at 37˚C and 5% $CO_2$ for three days. Cells were then collected through centrifugation at 1,000 x *g*

for 10 min and incubated in 20 μL of MTT dye (Sigma Aldrich, USA) at a concentration of 5 mg/mL in phosphate buffered saline (pH 7.2) at 37˚C and 5% $CO_2$. Following four hours of incubation with MTT dye, 150 μL of dimethylsulfoxide (DMSO, Sigma) was added to all samples to dissolve the formazan crystals. The optical density (OD) of the resulting solution was measured at a test wavelength of 490 nm and a reference wavelength of 630 nm using an ELISA microplate reader (MultiscanGo, Thermo Fisher Scientific, USA). Half maximal inhibitory concentration ($IC_{50}$) values were calculated for each cell line using GraphPad Prism 5 software (GraphPad Software, La Jolla, CA, USA).

## Mini- and micro-satellite genotyping of isolates

Parasite genetic diversity was assessed for schizont-infected cell lines showing high (A10/AT3 and A21/AT4), medium (A16/AT1, G3/BT) and low (A9/BT, N3/BT) $IC_{50}$ values (S1 Table). For this purpose, 1 x $10^6$ cells/mL were collected through centrifugation at 1,000 x *g* for 10 min and DNA prepared using the Promega Wizard genomic DNA extraction kit (Madison, WI, USA) following the manufacturer's instructions. Extracted DNA was resuspended in 100 μL rehydration buffer and stored at −20˚C. All DNA samples were then genotyped using parasite mini- and micro-satellite markers (TS5, 6, 8, 9, 12, 15, 16, 20 and 25) as described [26], although PCR primers used in the present study were not fluorescently tagged.

## Cloning and sequencing of *Cyto b* and *TaPIN1* genes of *T. annulata*

Drug resistant cell lines with high (A10/AT3 and A21/AT4), medium (A16/AT1, G3/BT) and low (A9/BT, N3/BT) $IC_{50}$ values were cloned by limiting dilution using the method described by Shiels *et al.* [27]. Putative clonal cell lines were further genotyped to validate that each clone represented a single parasite genotype. A minimum of ten clones from each isolate was then analysed by MTT assay to determine $IC_{50}$ values for the drug. DNA was extracted from two clones with the highest $IC_{50}$ value (Table 1) for each cell line. The DNA was then used in three independent PCRs to amplify a 1,089 bp region of the *Cyto b* gene of *T. annulata*. Primers used were: forward (5'-ATG AAT TTG TTT AAC TCA CAT TTG C- 3') and reverse (5'-

**Table 1. $IC_{50}$ values of clonal cell lines derived from A9/BT, A10/AT3, A16/AT1, A21/AT4, G3/BT and N3/BT isolates treated with buparvaquone.**

| Clone ID | $IC_{50}$ values (ng/mL) | | | | | |
|---|---|---|---|---|---|---|
| | A9/BT[b] | A10/AT3[a] | A16/AT1[b] | A21/AT4[a] | G3/BT[b] | N3/BT[b] |
| 1 | 0.86 | 13.89 | 1.85 | 16.75 | 2.89 | 3.28 |
| 2 | 0.91 | 29.78 | 1.61 | 25.53 | **9.21** * | 0.18 |
| 3 | 0.89 | 25.94 | 0.71 | **125.70** * | 1.28 | 0.95 |
| 4 | 1.15 | 16.99 | 0.75 | 6.36 | 7.48 | **14.19** * |
| 5 | **2.12** * | 12,62 | 2.70 | 63.72 | 3.19 | **10.70** * |
| 6 | **1.67** * | 47.37 | 1.90 | 18.95 | 2.61 | 2.31 |
| 7 | 0.69 | 77.74 | 1.80 | 13.33 | 2.69 | 9.66 |
| 8 | 0.50 | **99.85** * | 1.81 | **209.10** * | 0.62 | 1.05 |
| 9 | 1.34 | **137.70** * | **3.21** * | 30.66 | 0.92 | 6.20 |
| 10 | 0.78 | 63.95 | **4.66** * | 29.14 | **13.61** * | 0.51 |

(*) indicates clonal cell lines which were used for sequence analysis.

([a, b]) Different lower case superscript letters indicate statistical significance ($P < 0.05$). $IC_{50}$ values obtained from the clones of the cell lines A10/AT3 and A21/AT4 was significantly different from $IC_{50}$ values of the clones of the cell lines A9/BT, A16/AT1, G3/BT and N3/BT. There was no significant difference in $IC_{50}$ values between A10/AT3 and A21/AT4, and among the A9/BT, A16/AT1, G3/BT and N3/BT ($P > 0.05$).

TGC ACG AAC TCT TGC AGA GTC- 3'). These were designed to specifically amplify the open reading frame (ORF) of the gene based on publicly available sequence data (GenBank accession no: XM949625). PCR was performed with 45 mM Tris–HCl (pH 8.8), 11 mM $(NH_4)_2SO_4$, 4.5 mM $MgCl_2$, 0.113 mg/mL BSA, 4.4 μM EDTA, 1 nM dNTPs, 10 μM of each primer, 1 U *Taq* DNA polymerase (Solis BioDyne, Estonia) and 40 ng template DNA in a total volume of 50 μL using a Techne TC-512 thermocycler (Techne, UK) with the following conditions: initial denaturation at 94˚C for 5 min, followed by 30 cycles, denaturation at 95˚C for 1 min, annealing at 52˚C for 1 min and extension at 72˚C for 1 min 20 s. A final extension at 72˚C for 10 min was also included.

The *Cyto b* gene sequence of each clone was screened for the presence of any mutations corresponding with putative $Q_{O1}$ and $Q_{O2}$ domains of the encoded protein, located around the known ubiquinol binding site. Products of three independent *Cyto b* PCRs for each clone were separated on a 2% agarose gel and amplicons corresponding to a 1,089 bp product were excised from the gel using a QIAquick gel extraction kit (QIAGEN, Germany). Purified products were then cloned into the pCR4–TOPO plasmid vector (Invitrogen, Germany) using the TOPO® TA Cloning Kit for sequencing (Invitrogen, Germany). Three colonies were chosen for sequence analysis of each amplicon. Colonies were grown overnight and plasmids purified using the QIAGEN plasmid purification kit (QIAGEN, Germany).

The *TaPIN1* gene was screened for the presence of the mutation (A53P) at amino acid residue 53, within the catalytic loop of the predicted protein. For this purpose, four resistant clonal cell lines (A10/AT3/cl8 and A10/AT3/cl9 with the V135A mutation in *Cyto b* and A21/AT4/cl3 and A21/AT4/cl8 with the *Cyto b* P253S mutation) and three buparvaquone sensitive (A9/BT/cl5, G3/BT/cl2 and N3/BT/cl4) clonal cell lines were used. PCR amplification of a 900 bp region of the *TaPIN1* gene was performed as described by Marsolier *et al.* [22] using the following primers, Forward: 5'- GTC TGT CAA ATA GGT AGA AAT C- 3' and Reverse: 5'- GAG AGG AAG TTG AAT CAA ACA T- 3', as detailed above except that an annealing temperature of 56.6˚C was used. PCR amplicons of *TaPIN1* were cloned as described for the *Cyto b* gene.

Sequencing of *Cyto b* and *TaPIN1* genes was performed by a commercial sequencing service (Eurofins, Germany). All sequence data were deposited into GenBank database under the following accession numbers: MK693123 (A10/AT3/cl8), MK693124 (A10/AT3/cl9), MK693127 (A21/AT4/cl3), MK693126 (A21/AT4/cl8), MK693128 (A9/BT/cl5), MK693129 (A9/BT/cl6), MK693130 (N3/BT/cl4), MK693131 (N3/BT/cl5), MK693132 (G3/BT/cl2), MK693133 (G3/BT/cl10), MK693134 (A16/AT1/cl9) and MK693135 (A16/AT1/cl10).

## Allele-specific PCR assay

An allele-specific PCR (AS-PCR) was developed to determine the frequency of V135A and P253S mutations among the *T. annulata* isolates, clonal cell lines and blood samples from carrier animals used in the present study. When designing the primers, each single point mutation was positioned at the 3′ end of the forward primers, as previously indicated [28–31]. Two forward (one wild-type and one mutant) and one reverse (wild-type) primers were designed for each mutation. The optimum annealing temperatures were adjusted by performing a gradient PCR for each reaction. To detect the presence of the V135A mutation, PCR analyses were conducted using reverse primer V145A (5'-CCA TAA CCA GAT TGC ACT CCA-3') and one of the following allele-specific forward primers: Wild type 1-(Sensitive) (5'-GCT TCT GGG GAG CTA CAG T-3') and V135A-Resistant (5'-GCT TCT GGG GAG CTA CAG C-3'). For the P253S mutation, PCR analyses were performed using the reverse primer P253S (5'-CAA CAT GAA CAA CCA TCT TTC C-3') and one of the following allele-specific forward

primers: Wild type 2-(Sensitive) (5'-`CGA TAT TAT CTG ATC CTT TAA ACA CTC`-3') and P253S-Resistant (5'-`CGA TAT TAT CTG ATC CTT TAA ACA CTT`-3'). The 20 μL reaction mix consisted of 2 μL custom PCR mix (45 mM Tris-HCl, pH 8.8; 11 mM $(NH_4) SO_4$; 4.5mM $MgCl_2$; 0.113 mg/mL BSA; 4.4 mM EDTA), 1 μM of each dNTP (Thermo, UK), 10 μM of each primer, 2 μL of 1 U AmpliTaq DNA polymerase (Applied Biosystems, USA) and 40 ng of template DNA. PCR was carried out using a Techne TC-512 thermocycler (Techne, UK) with the following conditions: initial denaturation step at 94˚C for 3 min, followed by 30 cycles of amplification; 95˚C for 1 min, 62.5˚C for 1 min for the V135A mutation or 61˚C for 1 min for the P253S mutation annealing step and 72˚C for 1 min 20s extension, with a final extension of 72˚C for 10 min.

The sensitivity of the AS-PCR was determined by using diluted (0.2–20 ng/μL) DNA samples from resistant clonal cell lines. DNA samples from cloned lines possessing either the V135A (R1) or P253S (R2) mutations were mixed with DNA from a wild-type sensitive clonal cell line (S) (R1/S or R2/S). To test the specificity of the AS-PCR, DNA samples obtained from lines with V135A (R1) or P253S (R2) mutations were diluted (0.2–20 ng/μL) with each other (R1/R2: 1/10, 1/20; 1/50, 1/100 or R2/R1: 1/10, 1/20; 1/50, 1/100). PCR assays were performed as described above.

DNA was extracted from EDTA blood samples collected from carrier animals using the Promega Wizard genomic DNA extraction kit (Madison, WI, USA), following the manufacturer's instructions. Extracted DNA was resuspended in 100 μL rehydration buffer and stored at −20˚C. The DNA samples were then screened for *Cyto b* mutations using the AS-PCR protocol.

## Data analyses

Numerical values were first tested for normality using the Shapiro-Wilk test. The prevalence of mutations between years was compared using a Chi-Square or Fisher's exact test. $IC_{50}$ values that did not pass the normality test were analysed using the Kruskal–Wallis test followed by Dunn's test, with a Bonferroni adjustment for multiple comparisons. A value of $P < 0.05$ was considered significantly different. Statistical analyses were performed using SPSS software for Windows (Version 25.0. Armonk, NY: IBM Corp., USA).

## Ethics statement

The study was approved by the institutional animal Ethics Committee of the Aydın Adnan Menderes University (ethical identification number 050.04/2010/080) and conducted according to national guidelines and conforming to European Directive 2010/63/EU. All blood samples were obtained from privately owned farms. Before taking any blood samples from sick or healthy cattle, the owners of the cattle were verbally informed about the project and were then asked for their consent. All participants provided a written, informed consent indicating that they took part in the study voluntarily.

## Results

### Proliferative index of *T. annulata* isolates under drug pressure

The results of the MTT analyses conducted to determine the susceptibility of infected cell lines to buparvaquone demonstrated that these *T. annulata* field isolates showed a variety of phenotypes (S1 Table). Based on obtained $IC_{50}$ values, the isolates were divided into three groups representing low (1–3 ng/mL), moderate (3–7 ng/mL) and high (>7 ng/mL) resistance to buparvaquone. The majority of cell lines (94/140) were in the low $IC_{50}$ group, while the medium and high $IC_{50}$ groups comprised 38 and 8 isolates, respectively. The dose response

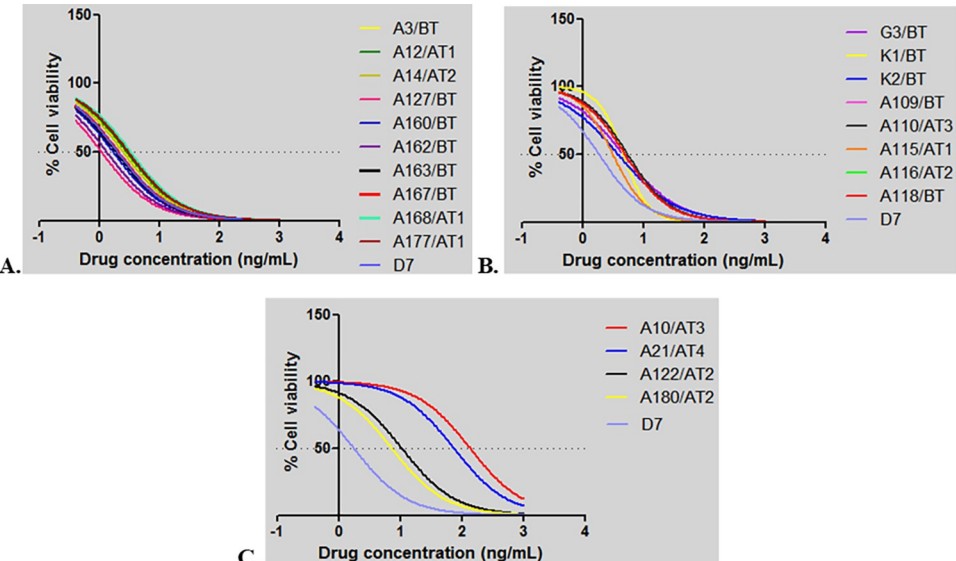

**Fig 1. MTT assay of *T. annulata* infected cell lines. A.** *T. annulata* isolates with low IC$_{50}$ values (2–3 ng/mL). **B.** *T. annulata* isolates with medium IC$_{50}$ values (3–7 ng/mL). **C.** *T. annulata* isolates with IC$_{50}$ values over (7 ng/mL). Numbers on the x axis (-1, 0, 1, 2, 3 and 4) indicates drug concentrations used to treat cell lines at doses of 0, 0.8, 7, 125, 1000 and 1500 ng/mL, respectively.

curves of a subset of isolates representing each group are shown in Fig 1. Evidence for selection of resistant parasites was found as, compared to the first drug treatment, a marked (five-fold) increase in the IC$_{50}$ values of isolates from animal A21 that received repeated buparvaquone treatment was observed over time. Thus, the IC$_{50}$ value of the isolates obtained from animal A21 after the first (A21/AT1) compared to the fourth (A21/AT4) buparvaquone injections were 16.75 ng/mL and 73.79 ng/mL, respectively (Fig 2B, S1 Table). Similarly, the IC$_{50}$ values of isolates obtained from animal A10 before (A10/BT) and after a third buparvaquone treatment (A10/AT3) were 16.75 ng/mL and 135 ng/mL, respectively (Fig 2A, S1 Table). This demonstrated a dramatic (almost ten-fold) increase in IC$_{50}$ values for infected cell lines isolated from animal A10 after repeated buparvaquone treatment.

## Evidence of genotypic selection in isolates after drug treatment

The genetic diversity of the parasites within the *T. annulata* schizont-infected cell lines with high (A10/AT3 and A21/AT4), medium (A16/AT1, G3/BT) and low (A9/BT, N3/BT) IC$_{50}$

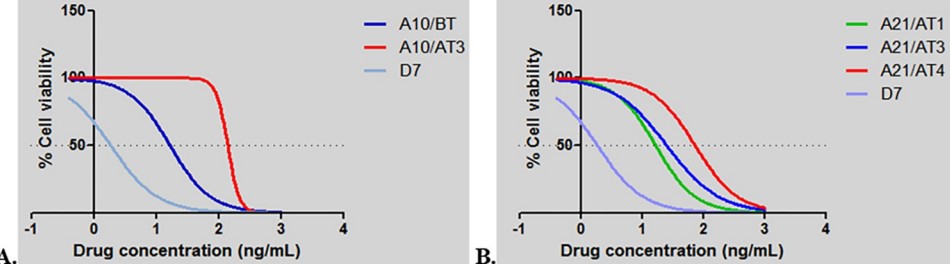

**Fig 2.** MTT assay of *T. annulata* infected cell lines obtained from animals A10 (A) and A21 (B) following repeated buparvaquone treatments. Numbers on the x axis (-1, 0, 1, 2, 3 and 4) indicates drug concentrations used to treat cell lines at doses of 0, 0.8, 7, 125, 1000 and 1500 ng/mL, respectively.

**Table 2. Number of alleles detected by nine micro- and mini-satellite markers before and after buparvaquone treatments.**

| Isolate ID | Mini- and micro-satellite markers | | | | | | | | |
|---|---|---|---|---|---|---|---|---|---|
| | TS5 | TS6 | TS8 | TS9 | TS12 | TS15 | TS16 | TS20 | TS25 |
| G3/BT | 1 | 2 | 2 | 2 | 2 | 1 | 1 | 3 | 3 |
| N3/BT | 1 | 1 | 2 | 3 | 1 | 1 | 1 | 1 | 2 |
| A9/BT | 2 | 2 | 2 | 3 | 2 | 2 | 2 | 2 | 3 |
| A16/AT1 | 2 | 1 | 1 | 1 | 2 | 1 | 1 | 1 | 1 |
| A10/BT | 2 | 2 | 1 | 2 | 2 | 2 | 1 | 2 | 2 |
| A10/AT3 | 1 | 1 | 1 | 1 | 1 | 1 | 1 | 1 | 1 |
| A21/AT1 | 6 | 4 | 6 | 2 | 2 | 2 | 1 | 1 | 4 |
| A21/AT3 | 2 | 1 | 2 | 2 | 2 | 1 | 1 | 1 | 2 |
| A21/AT4 | 2 | 1 | 1 | 1 | 1 | 1 | 1 | 1 | 2 |

**BT** indicates *T. annulata* isolates prepared before drug treatment.

**AT** indicates *T. annulata* isolates prepared after drug treatment. AT1-4 represents the number of drug treatments.

values was assessed by micro-satellite analysis for six of the lines. All cell lines, except for A10/AT3, comprised of a mixed parasite population (Table 2). Repeated-buparvaquone treatment reduced the diversity of the parasite population in lines isolated from animals A10 and A21. Only a single parasite genotype was detected following the third buparvaquone treatment in the cell line of the A10/AT3 isolate (Table 2).

Six cell lines from field isolates corresponding to the three different levels of buparvaquone resistance were cloned by limiting dilution and for each isolate, ten *bona fide* clones were selected. The presence of a single genotype in each clonal line was confirmed by multilocus genotyping. Results of MTT analyses performed with these clonal lines and $IC_{50}$ values are given in Table 1. The mean $IC_{50}$ values for the clones of isolates A10/AT3 and A21/AT4, both phenotypically resistant to buparvaquone, were 52.58 and 53.92 ng/mL, respectively. The mean $IC_{50}$ values of the clones of the isolates A9/BT, A16/AT1, G3/BT and N3/BT, all phenotypically sensitive to buparvaquone, were 1.09, 2.1, 4.45 and 4.90 ng/mL, respectively. Taken together the results indicate that treatment with buparvaquone has selected parasite genotypes that may harbour mutations conferring resistance to the drug.

## Sequence of *Cyto b* and *TaPIN1* genes of *T. annulata*

The *Cyto b* gene (1,089 bp) was sequenced for two representative clonal cell lines derived from each of the six isolates with high (A10/AT3 and A21/AT4), medium (A16/AT1, G3/BT) and low (A9/BT, N3/BT) $IC_{50}$ values. This was done in order to screen for the presence of non-synonymous mutations at the ubiquinone binding site, hypothesised to be associated with parasite resistance to buparvaquone. *Cyto b* sequences from the clonal cell lines were then aligned to and compared with the *Cyto b* gene (XM949625) sequence of the reference *T. annulata* (C9) genome. Six positions with non-synonymous mutations and six positions with synonymous nucleotide mutations were detected (Table 3). The six non-synonymous mutations sites were located at positions 151, 404, 436, 679, 757 and 1015 of the *Cyto b* gene, predicted to result in the following amino acid substitutions: M51L, V135A, A146T, V227M, P253S and A339V, respectively. Two of these non-synonymous mutations, between codons 116–144 ($Q_{o1}$) and 242–286 ($Q_{o2}$), respectively generate amino acid changes V135A and P253S located in the putative ubiquinone-binding sites ($Q_O$) [32,33]. Substitution V135A at the $Q_{o1}$ binding site was predicted in clones of A10/AT3, whereas substitution P253S at the $Q_{o2}$ binding site was predicted for clones of A21/AT4 (Fig 3).

**Table 3. Mutations detected in the *Cyto b* gene of *Tancytb* reference gene in PiroplasmDB database and different *T. annulata* isolates.**

| | Nucleotide | 151 | 234 | 348 | 404 | 417 | 429 | 436 | 576 | 679 | 757 | 870 | 1015 |
|---|---|---|---|---|---|---|---|---|---|---|---|---|---|
| | Codon | 51* | 78 | 116 | 135* | 139 | 143 | 146* | 192 | 227* | 253* | 290 | 339* |
| *Tancytb* (XM949625) | | ATG (Met) | TCG | ACT | GTC (Val) | TTA | TTC | GCT (Ala) | TCA | GTG (Val) | CCT (Pro) | GTA | GCT (Ala) |
| **G3/BT/cl2** (MK693132) | | … | ..A | ..C | … | ..G | ..T | A.. (Thr) | … | … | … | ..G | … |
| **G3/BT/cl10** (MK693133) | | … | ..A | ..C | … | ..G | ..T | A.. (Thr) | … | … | … | ..G | … |
| **N3/BT/cl4** (MK693130) | | … | ..A | ..C | … | ..G | ..T | A.. (Thr) | … | … | … | ..G | … |
| **N3/BT/cl5** (MK693131) | | … | ..A | ..C | … | ..G | ..T | A.. (Thr) | … | … | … | ..G | … |
| **A9/BT/cl5** (MK693128) | | … | ..A | ..C | … | ..G | ..T | A.. (Thr) | … | … | … | ..G | .T. (Val) |
| **A9/BT/cl6** (MK693129) | | … | ..A | ..C | … | ..G | ..T | A.. (Thr) | … | … | … | ..G | .T. (Val) |
| **A10/AT3/cl8** (MK693123) | | T.. (Leu) | ..A | … | .C. (Ala) | ..G | ..T | … | ..G | … | … | ..G | … |
| **A10/AT3/cl9** (MK693124) | | T.. (Leu) | ..A | … | .C. (Ala) | ..G | ..T | … | ..G | … | … | ..G | … |
| **A16/AT1/cl9** (MK693134) | | T.. (Leu) | ..A | … | … | ..G | ..T | … | ..G | … | … | ..G | … |
| **A16/AT1/cl10** (MK693135) | | T.. (Leu) | ..A | … | … | ..G | ..T | … | ..G | … | … | ..G | … |
| **A21/AT4/cl3** (MK693127) | | … | … | … | … | … | … | A.. (Thr) | … | A.. (Met) | T.. (Ser) | … | … |
| **A21/AT4/cl8** (MK693126) | | … | … | … | … | … | … | A.. (Thr) | … | A.. (Met) | T.. (Ser) | … | … |

(*) indicates the non-synonymous mutations sites.

Sequence analyses of the *TaPIN1* gene of the same twelve resistant and susceptible clonal cell lines demonstrated that the A53P substitution in the catalytic loop of the predicted protein reported by Marsolier *et al.* [22] was not present in any of the buparvaquone resistant and susceptible clonal cell lines examined (Fig 4, S2 Table). However, several other mutations in

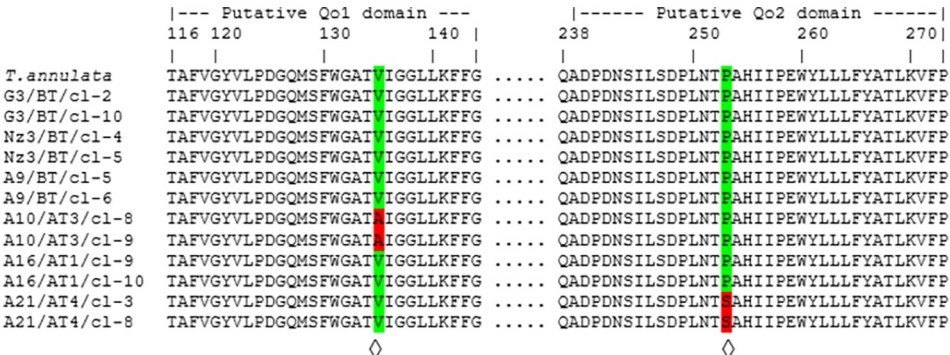

**Fig 3. Alignments of putative Q$_{o1}$ and Q$_{o2}$ domains of *Cyto b*.** Predicted amino acid sequences of Q$_{o1}$ and Q$_{o2}$ domains encoded by the *Cyto b* gene from the published *T. annulata* / C9 genome sequence and different *T. annulata* isolates with treatment failure history. Putative Q$_{o1}$ and Q$_{o2}$ domains are located between 116–144 and 238–273 amino acids of the protein around the ubiquinone binding site. Predicted amino acid substitutions associated with buparvaquone resistance are marked with ◊.

*TaPIN1* were detected which were predicted to result in amino acid substitution, including I2F, T22I, T22A, I23N, A26P, A26T, L78P and R96K. While the majority of the mutations were observed in sensitive clonal cell lines, three non-synonymous mutations (I2F, T22I and A26T) were detected only in the resistant clonal cell lines from A21/AT4 (clone 3 and clone 8). Additional mutations resulting in substitutions at amino acid positions 22 (T22A) and 26 (A26P) were also detected among the susceptible clonal cell lines.

## Development of AS-PCR to measure frequency of *Cyto b* V135A and P253S mutations in drug resistant *T. annulata* cell lines

The frequency of V135A and P253S substitutions detected at the putative drug binding domain of cytochrome b among resistant clonal cell lines (A10 and A21) was determined using an AS-PCR assay.

The developed AS-PCR assay was validated using diluted (0.2–20 ng/µL) DNA samples from lines with the V135A or P253S substitutions, demonstrated that 400 pg/µL DNA was sufficient to detect the targeted nucleotide mutations. Based on these data, clones of A9/BT (n = 45), A10/AT3 (n = 45), A16/AT1 (n = 40), A21/AT4 (n = 48), G3/BT (n = 9) and N3/BT (n = 42) isolates were analysed with the AS-PCR (S3 Table) to determine the presence of V135A and P253S mutations (S2 Fig). The mutation conferring V135A was detected in all 45 clones of isolate A10/AT3, while the mutation generating P253S was detected in all 48 clones of isolate A21/AT4. Following this result, corresponding pre-treatment (A10/BT) and early treatment (A21/AT1) isolates were cloned and sequenced to determine whether wild-type alleles were originally present and whether selection of resistant genotypes had occurred. AS-PCR analyses of clonal cell lines derived from these isolates revealed that while the mutation conferring V135A is present in the majority (38/41) of the clonal cell lines obtained from the isolate A10/BT, the mutation resulting in P253S is present only in 8 of 49 clonal cell lines obtained from isolate A21/AT1. For phenotypic characterisation, six clonal cell lines from isolates A10/BT and A21/AT1 (three with a mutation and three without any of the resistance associated mutations) were examined with the MTT assay. The results revealed that, compared to clones without the mutation (mean $IC_{50}$:1.19 ng/mL; range: 0.73–1.51 ng/mL), clones with V135A were resistant even at very high concentrations (mean $IC_{50}$: 45.28 ng/mL; range: 12.62–65.34 ng/mL) of buparvaquone (Table 4, Fig 5A). Similarly, the clones derived from isolate A21 with the P253S substitution had higher $IC_{50}$ values (mean $IC_{50}$: 21.0 ng/mL; range: 6.36–37.70 ng/mL) than clones without the associated nucleotide mutation (mean $IC_{50}$: 2.23 ng/mL; range: 1.97–3.17; Table 4, Fig 5B).

## Frequency of V135A and P253S mutations in field isolates

Having demonstrated a strong association between buparvaquone resistance and the presence of mutations in *Cyto b* that result in V135A and P253S substitutions, the AS-PCR developed for each mutation was used to measure the emergence and spread of drug resistance in the field. A total of 168 randomly selected *T. annulata*-positive blood samples and 140 *T. annulata* cell lines were tested. The AS-PCR results revealed that the V135A mutation was present in 10 of 168 carrier cattle (5.95%) and 2 of 140 (1.42%) isolates, whereas the P253S mutation was identified in 1 (0.59%) carrier animal and 10 (7.14%) isolates (Table 5). The frequency of mutations for V135A and P253S across all samples was 3.89% and 3.57%, respectively. It was apparent from the blood and cell line isolates examined in the two time periods, (A) 1998–2007 compared to (B) 2010–2011, that the frequency of the both mutations increased significantly in the latter period: V135A (P = 0.007), P253S (P = 0.000) (Table 5).

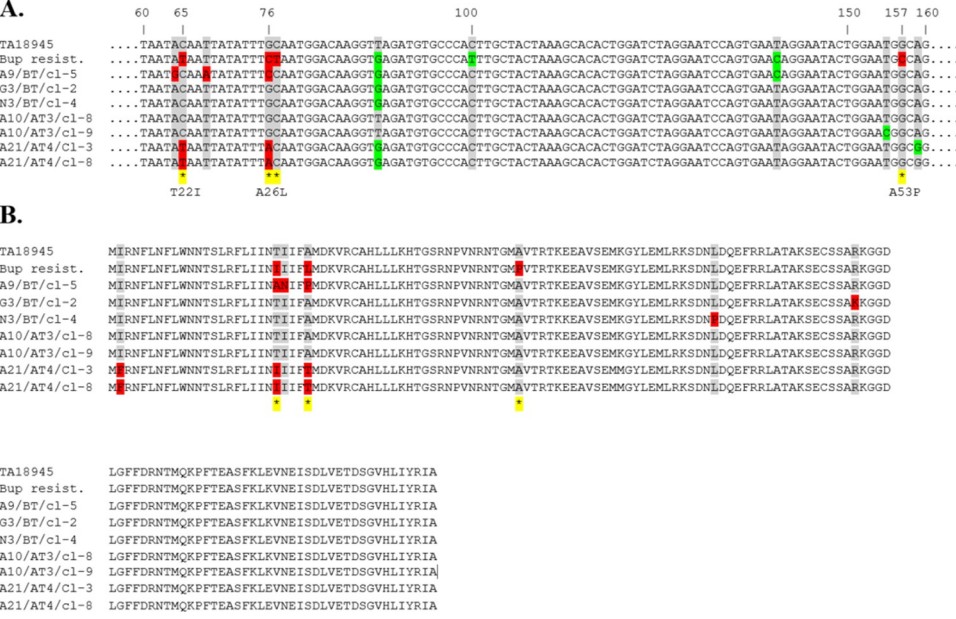

**Fig 4.** Alignments of *TaPIN1* nucleotide (A) and predicted amino acid (B) sequences. Sequences derived from a buparvaquone resistant Tunisian isolate and various Turkish *T. annulata* isolates with treatment failure history were compared for mutations with the *T. annulata* / C9 reference genome, obtained from PiroplasmDB.org database; the systematic identifier for *TaPIN1* is TA18945. Nonsynonymous substitutions detected among alignments were indicated with an asterisk (*) coloured in yellow.

## Discussion

The use of buparvaquone, a hydroxynaphthoquinone antiprotozoal drug akin to parvaquone and atovaquone, remains the only therapeutic option for the treatment of tropical theileriosis since the late 1980s [7]. However, several reports have indicated an increase in the number of treatment failures [8–11,13], raising the possibility of *Theileria annulata* developing resistance to buparvaquone. Buparvaquone resistance has been associated with mutations in the *Cyto b* gene [9–11,13]. However, it is important to note that in previous studies, the selection response of resistant parasite populations under drug pressure was not examined. This is crucial to provide evidence for a causal relationship between mutation(s) and resistance to the drug. Taking this into account in the present study, parasite isolates were characterised phenotypically (drug sensitivity) and genotypically (population structure/gene sequencing) to assess whether

**Table 4. IC$_{50}$ values of A10/BT and A21/AT1 clones with substitutions V135A and P253S.**

|  | V135A | | P253S | |
|---|---|---|---|---|
|  | **Clone code** | **IC$_{50}$ value (ng/mL)** | **Clone code** | **IC$_{50}$ value (ng/mL)** |
| **Clones with mutation** | A10/BT/clone-1 | 65.34 | A21/AT1/clone-13 | 6.36 |
|  | A10/BT/clone-2 | 57.90 | A21/AT1/clone-15 | 37.70 |
|  | A10/BT/clone-3 | 12.62 | A21/AT1/clone-17 | 18.95 |
| **Clones without mutation** | A10/BT/clone-7 | 0.73 | A21/AT1/clone-14 | 1.97 |
|  | A10/BT/clone-13 | 1.51 | A21/AT1/clone-24 | 1.55 |
|  | A10/BT/clone-21 | 1.33 | A21/AT1/clone-25 | 3.17 |

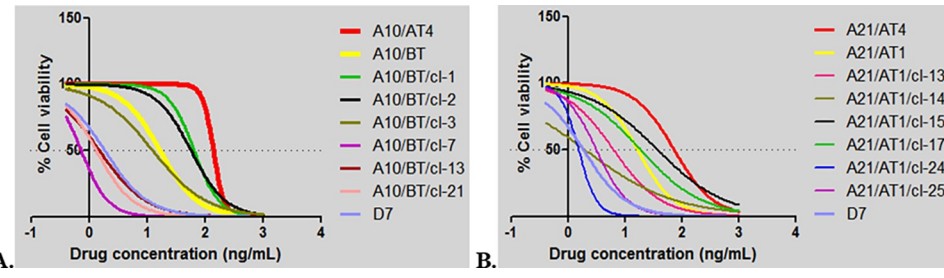

**Fig 5.** MTT assay of cloned cell lines from A10/BT (A) and A21/AT1 (B) isolates. Numbers on the x axis (-1, 0, 1, 2, 3 and 4) indicates drug concentrations used to treat cell lines at doses of 0, 0.8, 7, 125, 1000 and 1500 ng/mL, respectively.

treatment failure observed in infected cattle in the Aydın region of Türkiye is accompanied by selection of genotypes bearing mutations in *Cyto b* that are linked to buparvaquone resistance.

The results showed that, while the majority of the cell lines tested were sensitive to buparvaquone, those representing isolates obtained from animals which had received repeated-buparvaquone treatments displayed five to ten-fold higher $IC_{50}$ values. Hence, there is a clear correlation between repetitive buparvaquone treatment and evidence of emergence of infected cells resistant to the drug i*n vivo*. Furthermore, emergence of drug resistance was found to be associated with a marked reduction in the genetic complexity of the parasite population in cell lines displaying high $IC_{50}$ values (Table 2). This supports the conclusion that selection of parasite genotypes resistant to the drug has occurred in animals under a repeated treatment regime, and that infection in these animals is represented by only one or two genotypes. The level of parasite genetic diversity within drug resistant isolates is much lower than observed in isolates analysed in previous studies, focused on investigation the broader parasite population structure in Türkiye [6]. Furthermore, a reduction in genetic complexity through selection in the field can be predicted to enhance transmission resistant genotypes to the vector ticks, *Hyalomma* spp., and facilitate the spread of drug resistance in regions endemic for theileriosis. In support of this scenario, Bell *et al.* [34] demonstrated that treatment with pyrimethamine resulted in a shift in the genetic composition of *Plasmodium chabaudi* gametocyte populations in host venous blood towards resistant parasite populations and these were effectively transmitted to vector mosquitos. Further studies are warranted to investigate the rate at which drug resistance disseminates in *T. annulata* populations, taking account of vector transmission intensity and drug pressure. The increase in detection of stable mutations linked to drug resistance in field populations over time (Table 5) suggests tick transmission of resistance genotypes plays a significant role.

**Table 5. Frequency of V135A and P253S mutations in independently collected *T. annulata* positive blood samples and isolates.**

| | Number of samples tested | | Presence of Mutations | | | |
|---|---|---|---|---|---|---|
| | | | V135A | | P253S | |
| Time interval for sample collection (Year) | Blood/cell isolates (n) | Total | Blood/cell isolates | Total | Blood/cell isolates | Total |
| (A) 1998–2007 | 119/50 | 169 | 2 (1.68%) / 0 (0.0%) | 2 (1.19%) | 0 (0.0%) / 0 (0.0%) | 0 (0.0%) |
| (B) 2011–2012 | 49/90 | 139 | 8 (16.32%) / 2 (2.22%) | 10 (7.19%) | 1 (2.04%) /10 (11.11%) | 11 (7.91%) |
| **TOTAL** | 168/140 | 308 | 10 (5.95%) / 2 (1.42%) | 12 (3.89%) | 1 (0.59%) / 10 (7.14%) | 11 (3.57%) |
| **P value** | | | **0.001[*] / 0.292** | **0.007[*]** | **0.292 / 0.014[*]** | **0.000[*]** |

(\*) denotes significant difference in mutated allele frequency in period A compared to period B.

Sequence analyses of the *Cyto b* gene of clonal cell lines derived from *T. annulata* isolates with high, medium and low $IC_{50}$ values revealed that two of the six non-synonymous mutations, namely V135A and P253S, were located at the putative ubiquinone-binding sites between codons 116–144 ($Q_{o1}$) and 242–286 ($Q_{o2}$) of the predicted protein. Work on related parasites investigating drugs in the same hydroxynaphthoquinone family as buparvaquone highlighted mutations in these regions of *Cyto b* as conferring drug resistance [35–38].

A comparison of the *T. annulata Cyto b* gene sequence obtained in the present study with those obtained in studies conducted in Tunisia [9], Iran [10], Sudan [11] and Egypt [13] reveals similarities as well as differences that might prove important in elucidating the role of particular *Cyto b* mutations in conferring buparvaquone resistance (S4 Table). Firstly, the P253S mutation at the $Q_{o2}$ binding region was found to be common in all studies, except for the Sudanese investigation. The V227M mutation near the $Q_{o2}$ binding region detected in blood samples collected in Sudan was also observed in clones of the A21/AT4 resistant *T. annulata* isolate in the present study. The A146T mutation was detected in both resistant and sensitive cell lines in all Tunisian, Sudanese and Turkish isolates. The mutation S129G, detected in Tunisian, Sudanese and Iranian isolates, was not detected in the present study. Three synonymous mutations located at nucleotide positions 417, 429 and 870 (codons 139, 143 and 290) of the *Cyto b* gene were also reported in all studies except those from Egypt and Iran. One common non-synonymous mutation at codon 146 [9,11,13] was also detected in the present study in all samples except for A10 (resistance) and A16 (sensitive) isolates. Thus, the P253S substitution is the most geographically widespread and it may be hypothesised that it has arisen on multiple occasions. Alternatively, this mutation may have been the earliest to develop and has spread across endemic regions coincident with buparvaquone resistance.

Two lines of evidence support hypothesis that the V135A mutation is associated with resistance of *T. annulata* to buparvaquone: (i) similar to the P253S substitution, clonal infected cell lines with V135A have significantly higher $IC_{50}$ values than those without this mutation, (ii) heterogeneous parasite populations before first buparvaquone treatment (A10/BT in the case of V135A) became clonal, and exclusively buparvaquone-resistant parasite populations were isolated from animals receiving third and/or fourth buparvaquone treatments. We conclude that for the time period and region under study, mutations that result in V135A and P253S are most likely to confer and promote selection of buparvaquone resistant parasites.

Functional validation by reverse genetics of mutations in the *Cyto b* gene cannot be validated for *Theileria* and it should be appreciated that an alternative mechanism of buparvaquone resistance may operate or involve multiple mutations in *Cyto b*. Additional mechanisms could operate in isolation or increase the level of resistance to the drug in combination with *Cyto b* mutations. For example, *P. falciparum*-resistance to sulphadoxine-pyrimethamine (SP) is classified as "partially resistant", "fully resistant" and "super resistant" depending on the combination of mutations on two genes, namely dihydrofolate reductase (*Dhfr*) and dihydropteroate synthase [39].

A study by Marsolier *et al.* [22] suggested that a A53P mutation, in the catalytic loop of TaPIN1 is associated with resistance of *T. annulata* to buparvaquone. While the A53P mutation was not present in any of the resistant or susceptible clonal cell lines examined in the present study, several mutations (I2F, T22I, T22A, I23N, A26P, A26T and R96K) were detected. Of these, only mutations I2F, T22I, and A26T were identified in the resistant clonal cell lines obtained from A21/AT4 and these also possessed the *Cyto b* mutation P253S. Other mutations at residues 22 (T22A) and 26 (A26P) were observed in susceptible clonal cell lines, suggesting they do not confer resistance. Interestingly, mutations at codons 22 and 26 were also detected in both Tunisian [22] and Sudanese [12] isolates (S2 Table) although it is unknown whether ongoing selective pressure maintains these alleles in these areas. There was no example of a

*TaPIN1* mutant in the absence of a mutated *Cyto b* gene in any of the cell lines associated with a high level of buparvaquone resistance in our study, and for clones derived from the A10 isolate, mutations in *TaPIN1* were shown not to be required to confer drug resistance. The current study does not reveal any evidence of *TaPIN1* mutations being involved in resistance to buparvaquone.

To determine the allelic frequency of V135A and P253S in the field, an allele-specific PCR method was developed and deployed. AS-PCR is a relatively simple, inexpensive method and has been widely used in drug resistance studies for several parasites such as *Plasmodium falciparum* [40,41] and *Haemonchus contortus* [42]. Results obtained in the present study demonstrated that overall frequency of alleles corresponding to V135A and P253S was less than 4% (Table 5). It is important to note, however, that there was an almost ten-fold increase in the frequency of both alleles in samples analysed between 1998–2007 to 2011–2012, indicating a marked increase in the proportion of resistant *T. annulata* parasites in circulation in the latter period. These data could explain, at least in part, the increase in the number of cases with treatment failures using buparvaquone. It is also evident that unless better treatment and control strategies are developed, an increase in the frequency of resistant parasite populations is likely. It should be noted though, that while the mutations detected by the AS-PCR may still be dominant in current drug resistant populations, additional mutations could have emerged during the last ten years. There is a need, therefore, to reassess parasite populations for mutations causally linked to buparvaquone resistance, potentially on a regular basis. Nevertheless, the AS-PCR results highlight the value of monitoring temporospatial differences in drug resistant parasite populations in the field to provide an ongoing risk assessment of treatment failure in endemic regions.

In summary, data gathered in the present study strongly suggest that selection of *T. annulata* genotypes bearing mutations resulting in V135A and P253S substitutions at the putative binding sites of buparvaquone in cytochrome b play a key role in conferring resistance to buparvaquone. However, until gene replacement/mutation of the *Theileria* genome can be engineered, the possibility that mutations other than V135A and P253S of *Cyto b* and involvement of additional candidate genes such as *TaPIN1* cannot be fully excluded. Identification of resistance markers and development of AS-PCR to track drug resistant parasites should be valuable for monitoring the effectiveness of treatment strategies for control of tropical theileriosis.

## Supporting information

**S1 Fig. Geographical distribution of the parasite material used in the present study.** The geographical distribution of the provinces where the parasite material used in this study was obtained. The map was prepared using the USGS National Map Viewer (public domain): http://viewer.nationalmap.gov/viewer/ with some modification.
(TIF)

**S2 Fig. Agarose gel electrophoresis of AS-PCR.** Gels showing detection of PCR amplicons of isolates with mutations V135A (A) and P253S (B), respectively. Products amplified using drug sensitive and resistance specific forward primers are given at the top and bottom of each gel, respectively. M, 100 bp molecular size marker (Thermo Scientific Corp.); lanes 1–16, products from template DNA of *T. annulata* samples; lanes S sensitive control, lanes R resistance control, lane X mixed control.
(TIF)

**S1 Table. IC$_{50}$ values of different *T. annulata* isolates prepared before and after buparvaquone treatment.**
(PDF)

**S2 Table. Summary of mutations detected in *TaPIN1* sequences of *T. annulata* isolates from Tunisia, Sudan and Turkey.**
(PDF)

**S3 Table. AS-PCR results of clonal cell lines.**
(PDF)

**S4 Table. Summary of mutations detected in *Cyto b* sequences of *T. annulata* isolates from Tunisia, Sudan, Iran and Turkey.**
(PDF)

## Author Contributions

**Conceptualization:** Selin Hacılarlıoglu, Andrew Tait, Brian Shiels, Tulin Karagenc.

**Data curation:** Selin Hacılarlıoglu, Tulin Karagenc.

**Formal analysis:** Serkan Bakırcı.

**Funding acquisition:** Andrew Tait, Tulin Karagenc.

**Investigation:** Selin Hacılarlıoglu, Huseyin Bilgin Bilgic, Serkan Bakırcı, Andrew Tait, William Weir, Brian Shiels, Tulin Karagenc.

**Methodology:** Selin Hacılarlıoglu, Huseyin Bilgin Bilgic, Andrew Tait, William Weir, Brian Shiels, Tulin Karagenc.

**Project administration:** Andrew Tait, Tulin Karagenc.

**Resources:** Huseyin Bilgin Bilgic, Andrew Tait, Brian Shiels, Tulin Karagenc.

**Supervision:** Andrew Tait, Tulin Karagenc.

**Validation:** Selin Hacılarlıoglu, Tulin Karagenc.

**Visualization:** Selin Hacılarlıoglu, William Weir.

**Writing – original draft:** Selin Hacılarlıoglu.

**Writing – review & editing:** Huseyin Bilgin Bilgic, Serkan Bakırcı, Andrew Tait, William Weir, Brian Shiels, Tulin Karagenc.

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
