## [Decision Letter · Decision Letter 0]

23 Aug 2022

PONE-D-22-20200Mutations in Theileria annulata cytochrome b gene associated with resistance to buparvaquonePLOS ONE

Dear Dr. Karagenc,

Thank you for submitting your manuscript to PLOS ONE. After careful consideration, we feel that it has merit but does not fully meet PLOS ONE’s publication criteria as it currently stands. Therefore, we invite you to submit a revised version of the manuscript that addresses the points raised during the review process.

We look forward to receiving your revised manuscript.

Kind regards,

Vikrant Sudan, PhD

Academic Editor

PLOS ONE

Journal Requirements:

5. We note that Figure S1 in your submission contain [map/satellite] images which may be copyrighted. All PLOS content is published under the Creative Commons Attribution License (CC BY 4.0), which means that the manuscript, images, and Supporting Information files will be freely available online, and any third party is permitted to access, download, copy, distribute, and use these materials in any way, even commercially, with proper attribution. For these reasons, we cannot publish previously copyrighted maps or satellite images created using proprietary data, such as Google software (Google Maps, Street View, and Earth). For more information, see our copyright guidelines: http://journals.plos.org/plosone/s/licenses-and-copyright.

a. You may seek permission from the original copyright holder of Figure S1 to publish the content specifically under the CC BY 4.0 license.  

Additional Editor Comments:

I recommend major revisions.

Kindly go through the individual point wise concern raised by both the reviewers.

While revision the manuscript kind go through each comment and respond accordingly.

Reviewers' comments:

Reviewer's Responses to Questions

**Comments to the Author**

1. Is the manuscript technically sound, and do the data support the conclusions?

Reviewer #1: Yes

Reviewer #2: Yes

2. Has the statistical analysis been performed appropriately and rigorously? 

Reviewer #1: No

Reviewer #2: Yes

3. Have the authors made all data underlying the findings in their manuscript fully available?

Reviewer #1: Yes

Reviewer #2: Yes

4. Is the manuscript presented in an intelligible fashion and written in standard English?

Reviewer #1: Yes

Reviewer #2: Yes

5. Review Comments to the Author

Reviewer #1: The manuscript entitled ‘Mutations in Theileria annulata cytochrome b gene associated with resistance to buparvaquone’ is well written. The study is well designed, but lacks novelty. The following suggestions need to incorporate in the manuscript;

1. The mutations V135A and P253S at the putative binding sites of buparvaquone in Cyto b is not a new finding. So many authors have already reported about the same from different parts of the world. The authors need to specify the novelty of the work in the manuscript. Whether it is in vitro or field study, the mutations remains in the resistant parasites.

2. Further the study was conducted during 1998 and 2012 and a lot of new mutations might have emerged in the putative drug binding sites during the last 10 years (upto 2022). The authors need to discuss about it in the discussion.

3. In Line number 20: How the sample size of 16 was arrived? Which sampling technique was used in the present study? The sampling method need to be mentioned clearly.

Minor corrections:

4. In line number 105: A full stop needs to be kept at the last of the sentence.

5. Line 165; The word’ available’ need to be deleted as it is repeated.

6. Line Number 166: ‘PCR conditions comprised’ need to be replaced with ‘PCR was standardized with’

7. Line Number 168 and 224; The concentration of template DNA in terms of ng need to be mentioned not as volume.

8. Line Number 211; The bracket need to be uniform. Avoid using one square bracket

Reviewer #2: Comments on the manuscript “Mutations in Theileria annulata cytochrome b gene associated with resistance to buparvaquone” (PONE-D-22-20200)

The manuscript “Mutations in Theileria annulata cytochrome b gene associated with resistance to buparvaquone” describes the mutations observed in the Cytochrome b gene of Theileria annulata on exposure to buparvaquone and its relationship with resistance against this drug. The study is novel and performed systematically. However, in many places, the language is used improperly. Hence it needs thorough language editing.

Discussion: Too lengthy. May be reduced to 3/4th. Many generalizations may be avoided. Repetition of results is to be avoided.

References: Strictly follow the journal style. Many scientific names are not italicized. Many titles of articles are written with the capitalization of each word.

Minor points

Line 49: Is it correct to write “resistance, was not detected in any of the resistant and susceptible clonal cell lines”?

Line 76: indicated

Line 79: Remove also

Line 94: Remove particular

Line 109: were

Line 113 and other places: Better to write “is shown in supporting information (Fig S1). Correct other places also.

Line 126,127 and many other places: ml should be replaced by mL.

Line 135: Revise writing “drug sensitive control parasite population”

Line 165: Remove “available” which is repeating

Line 168: Taq should be italicized

Line 147, 156: It is difficult for a naïve reader to understand what is A10/AT3. In the “parasite material” section you may add another paragraph to reveal how you have named the isolates.

Line 192: sequencing need not be capitalized for the first letter

Line 204-206: This is part of the results. Rewrite

Line 209-211: the primer design is not clear from the sentence written

Line 211: primers

Line 246,298: SPSS/MTT: Do not begin a sentence with an abbreviation

Line 255: observed/calculated instead of obtained

Line 263-265: rewrite.

Line 270,275: Figure headings should be changed

Line 284: Macroschizont is a parasite? The sentence may be rewritten. Also, whether the macro/microschizont concepts are still there?

Line 292,372,375,390: Remove full stop after heading

Line 299: Remove parenthesis

Line 364: “obtained” may be changed

Line 415: “These results indicate that” may be replaced with Hence,

Line 427-429: Rewrite the sentence

Line 441: demonstrated

Line 452- 463, 491-512: The information may be tabulated.

6. PLOS authors have the option to publish the peer review history of their article (what does this mean?). If published, this will include your full peer review and any attached files.

Reviewer #1: **Yes: **Dr. Siju Susan Jacob

Reviewer #2: **Yes: **REGHU RAVINDRAN

---

## [Author Response · Author response to Decision Letter 0]

9 Nov 2022

Dear Dr. Vikrant Sudan, 

Thank you for your response to our manuscript PONE-D-22-20200 entitled "PONE-D-22-20200 Mutations in Theileria annulata cytochrome b gene associated with resistance to buparvaquone". We would also like to thank both the reviewers for providing specific comments and suggestions to improve the quality of the manuscript.

On the following pages, we present our point-by-point responses to all the comments and suggestions made by the Academic Editor and reviewers, including a proposed title change, “Selection of genotypes harbouring mutations in the cytochrome b gene of Theileria annulata is associated with resistance to buparvaquone”. A revised version of our manuscript is also provided, where all modifications are shown in a separate file labeled 'Revised Manuscript with Track Changes’. Please be informed that all the changes made in the text as a response to reviewers are underlined and highlighted in yellow collor. 

Yours sincerely,

Tulin KARAGENC (DVM, PhD)

 

Response to Journal Requirements:

Comment 1. Please ensure that your manuscript meets PLOS ONE's style requirements, including those for file naming. The PLOS ONE style templates can be found at

The manuscript has been re-formatted according to PLOS ONE's style requirements, including the naming of files.

Comment 2. In your Methods section, please provide additional information regarding the permits you obtained for the work. Please ensure you have included the full name of the authority that approved the field site access and, if no permits were required, a brief statement explaining why.

Ethics statement has been changed as below: 

 “The study was approved by the institutional animal Ethics Committee of the Aydın Adnan Menderes University (ethical identification number 050.04/2010/080) and conducted according to national guidelines and conforming to European Directive 2010/63/EU. All blood samples were obtained from privately owned farms. Before taking any blood samples from sick or healthy cattle, the owners of the cattle were verbally informed about the project and were then asked for their consent. All participants provided a written, informed consent indicating that they took part in the study voluntarily.” (Lines 259-265 in the modified manuscript).

Comment 3. Please provide additional details regarding participant consent. In the ethics statement in the Methods and online submission information, please ensure that you have specified what type you obtained (for instance, written or verbal, and if verbal, how it was documented and witnessed). If your study included minors, state whether you obtained consent from parents or guardians. If the need for consent was waived by the ethics committee, please include this information.

Both verbal and written consents of participants were obtained as indicated above in Response under ethics statement.

Comment 4. We note that the grant information you provided in the ‘Funding Information’ and ‘Financial Disclosure’ sections do not match.

Grant number is provided in the “Funding Information” section instead of “Financial Disclosure” section. 

Comment 5. We note that Figure S1 in your submission contain [map/satellite] images which may be copyrighted. All PLOS content is published under the Creative Commons Attribution License (CC BY 4.0), which means that the manuscript, images, and Supporting Information files will be freely available online, and any third party is permitted to access, download, copy, distribute, and use these materials in any way, even commercially, with proper attribution. For these reasons, we cannot publish previously copyrighted maps or satellite images created using proprietary data, such as Google software (Google Maps, Street View, and Earth). For more information, see our copyright guidelines: http://journals.plos.org/plosone/s/licenses-and-copyright.

Figure S1 was changed using the USGS National Map Viewer (public domain): http://viewer.nationalmap.gov/viewer/. If the quality of the image is not satisfactory to the journal’s standards, we can consider removing it all together. The legend of the S1 Fig has been changed to “S1 Fig. Geographical distribution of the parasite material used in the present study. The geographical distribution of the provinces where the parasite material used in this study was obtained. The map was prepared using the USGS National Map Viewer (public domain): http://viewer.nationalmap.gov/viewer/ with some modification.” (Lines 673-676 in the modified manuscript).

Response to Reviewers' comments: 

Response to Reviewer 1:

Reviewer #1: The manuscript entitled ‘Mutations in Theileria annulata cytochrome b gene associated with resistance to buparvaquone’ is well written. The study is well designed, but lacks novelty. The following suggestions need to incorporate in the manuscript;

Comment 1. The mutations V135A and P253S at the putative binding sites of buparvaquone in Cyto b is not a new finding. So many authors have already reported about the same from different parts of the world. The authors need to specify the novelty of the work in the manuscript.

Whether it is in vitro or field study, the mutations remains in the resistant parasites.

The three main novel features of the manuscript are [1] characterisation of parasite populations both phenotypically (MTT assay) and genotypically (sequencing, mini- and micro satellite analyses) based on their response to buparvaquone treatment at various doses, [2] the development of a simple AS-PCR assay to determine resistant parasite populations in the field and [3] directly demonstrating selection of parasite genotypes in response to drug pressure. The first is relevant, as it provides evidence of a causal relationship between mutation (genotype) and resistance phenotype (rather than a simple association of previous studies) and provides indication on the rapidity of selection of a resistant parasite population from a mixed population; the second is important as it provides a simple test to screen parasite populations for genotypes harbouring mutations associated with drug resistance, providing a tool to monitor the spread of resistance in the field. The third finding is particularly novel and important and for this reason we suggest modifying the title to “Selection of genotypes harbouring mutations in the cytochrome b gene of Theileria annulata is associated with resistance to buparvaquone”. Consequently, the following paragraphs of the discussion are updated: 

“Buparvaquone resistance has been associated with mutations in the Cyto b gene [9-11,13]. However, it is important to note that in previous studies, the selection response of resistant parasite populations under drug pressure was not examined. This is crucial to provide evidence for a causal relationship between mutation(s) and resistance to the drug. Taking this into account in the present study, parasite isolates were characterised phenotypically (drug sensitivity) and genotypically (population structure/gene sequencing) to assess whether treatment failure observed in infected cattle in the Aydın region of Türkiye is accompanied by selection of genotypes bearing mutations in Cyto b that are linked to buparvaquone resistance.” (Lines 415-422 in the modified manuscript).

“In summary, data gathered in the present study strongly suggest that selection of T. annulata genotypes bearing mutations resulting in V135A and P253S substitutions at the putative binding sites of buparvaquone in cytochrome b play a key role in conferring resistance to buparvaquone. However, until gene replacement/mutation of the Theileria genome can be engineered, the possibility that mutations other than V135A and P253S of Cyto b and involvement of additional candidate genes such as TaPIN1 cannot be fully excluded. Identification of resistance markers and development of AS-PCR to track drug resistant parasites should be valuable for monitoring the effectiveness of treatment strategies for control of tropical theileriosis.” (Lines 517-525 in the modified manuscript).

In addition, while data has indicated an association of mutant TaPIN1 genes with buparvaquone resistance, they are not always present in resistant lines. Thus, there is a requirement for further analysis to further establish the strength of association between TaPIN1 and drug resistance and its importance relative to mutations in Cyto b. This was performed and results discussed in the current study between the lines 484-498 in the modified manuscript.

Comment 2. Further the study was conducted during 1998 and 2012 and a lot of new mutations might have emerged in the putative drug binding sites during the last 10 years (upto 2022). The authors need to discuss about it in the discussion.

We agree with the point that new mutations might have been emerged during the last ten years. This issue was addressed in the discussion (Lines 508-516 in the modified manuscript) as follows:

“It is also evident that unless better treatment and control strategies are developed, an increase in the frequency of resistant parasite populations is likely. It should be noted though, that while the mutations detected by the AS-PCR may still be dominant in current drug resistant populations, additional mutations could have emerged during the last ten years. There is a need, therefore, to reassess parasite populations for mutations causally linked to buparvaquone resistance, potentially on a regular basis. Nevertheless, the AS-PCR results highlight the value of monitoring temporospatial differences in drug resistant parasite populations in the field to provide an ongoing risk assessment of treatment failure in endemic regions.”

Comment 3. In Line number 20: How the sample size of 16 was arrived? Which sampling technique was used in the present study? The sampling method need to be mentioned clearly.

Whole blood samples were obtained from a total of more than five hundred, randomly selected healthy cattle in the study area described in the manuscript. Of these, a total of 168 were found to be positive for T. annulata by PCR. To clarify the issues of the number of samples used in the present study, the “parasite material” section in Material and Methods was changed as follows (Lines 105-130 in the modified manuscript):

“A total of 140 T. annulata schizont-infected cell lines obtained from animals in the acute phase of theileriosis and 168 T. annulata piroplasm-positive blood samples obtained from healthy carrier animals were used in the present study. These samples were gathered between 1998 and 2012 from different farms located within nine different provinces (Center, Söke, Germencik, Kocarlı, İncirliova, Cine, Akçaova, Kösk and Nazilli) of Aydın in Western Türkiye, where tropical theileriosis is endemic.

In order to obtain T. annulata schizont infected cell lines, 10 mL blood samples were collected into heparinised tubes from cattle showing clinical signs of the disease before and/or after buparvaquone treatment(s). In vitro establishment of schizont-infected cells from peripheral mononuclear cells (PBM) was carried out as previously described [23]. Once the cell line established, a code was given to each isolate according to the site where the blood samples were taken. The letters M, S, G, CN, AC, C, A, K and N stand for the provinces of Center, Söke, Germencik, Kocarlı, İncirliova, Cine, Akçaova, Kösk and Nazilli, respectively. The numbers following these letters indicate the identification number of the animal from which the T. annulata schizont cell culture was established, viz. A10 stands for the 10th animal from the Akçaova province. This was followed by the timing (BT: before or AT: after treatment) and the number of buparvaquone treatment. Accordingly, A10/BT stands for the isolate obtained from the 10th animal in Akçaova province before the buparvaquone treatment. Similarly, A21/AT4 stands for the isolate obtained from the 21st animal in Akçaova province after the 4th buparvaquone treatment.

In an attempt to determine the frequency of resistant parasites in Aydın region, whole blood samples were obtained from a total of more than five hundred, randomly selected healthy cattle in the selected sampling sites. Of these, a total of 168 were found to be positive for T. annulata by PCR using Cyto b gene primers as described previously [6].

A map illustrating the geographical location of the sampling sites is shown in supporting information (S1 Fig) with isolate details summarised in S1 Table.”

Minor corrections:

Comment 4. In line number 105: A full stop needs to be kept at the last of the sentence. 

A full stop has been put at the end of the sentence. However the sentence “Sequence analyses of TaPIN1 gene were also performed to examine if there is any association of A53P mutation with resistance of the infected lines to buparvaquone” has been deleted while the manuscript has been comprehensively re-edited. 

Comment 5. Line 165; The word’ available’ need to be deleted as it is repeated. 

The extra “available” has been deleted. (Line 175 in the modified manuscript).

Comment 6. Line Number 166: ‘PCR conditions comprised’ need to be replaced with ‘PCR was standardized with’ 

 ‘PCR conditions comprised’ was replaced by “PCR was performed with-----”. (Line 176 in the modified manuscript).

Comment 7. Line Number 168 and 224; The concentration of template DNA in terms of ng need to be mentioned not as volume. 

The concentration of template DNA has been corrected as “40 ng template DNA “ (Lines 178 and 233 in the modified manuscript).

Comment 8. Line Number 211; The bracket need to be uniform. Avoid using one square bracket

The format of brackets has been corrected (Line 220 in the modified manuscript).

Response to Reviewer 2:

Reviewer #2: Comments on the manuscript “Mutations in Theileria annulata cytochrome b gene associated with resistance to buparvaquone” (PONE-D-22-20200)

The manuscript “Mutations in Theileria annulata cytochrome b gene associated with resistance to buparvaquone” describes the mutations observed in the Cytochrome b gene of Theileria annulata on exposure to buparvaquone and its relationship with resistance against this drug. The study is novel and performed systematically. However, in many places, the language is used improperly. Hence it needs thorough language editing.

In order to improve the language of the manuscript, it has been comprehensively re-edited by our co-authors Professors Brian Shiels and William Weir, both native speakers of English; extensive edits have been made throughout.

Comment. Discussion: Too lengthy. May be reduced to 3/4th. Many generalizations may be avoided. Repetition of results is to be avoided.

The discussion has been substantially reduced and repetitive sections removed. 

Comment. References: Strictly follow the journal style. Many scientific names are not italicized. Many titles of articles are written with the capitalization of each word.

All the references have been checked and have been corrected according to the journal style.

Minor points

Comment. Line 49: Is it correct to write “resistance, was not detected in any of the resistant and susceptible clonal cell lines”?

The sentence has been re-worded as follows: “The A53P mutation of TaPIN1 of T. annulata, previously suggested as being involved in buparvaquone resistance, was not detected in any of the clonal cell lines examined in the present study.” (Lines 45-47 in the modified manuscript).

Comment. Line 76: indicated 

The word “indicate” has been replaced by “demonstrated” (Line 74 in the modified manuscript).

Comment. Line 79: Remove also

The word “also” has been removed. The sentence beginning with “Therefore, it was logical to propose….” has been re-worded as “Based on the structural similarities between buparvaquone and atovaquone, it was logical to propose that a mutation in Cyto b of T. annulata, particularly at the binding site of ubiquinone, would be associated with resistance to buparvaquone [8- 11].” (Lines 75-78 in the modified manuscript).

Comment. Line 94: Remove particular

The word “particular” has been removed. The sentence beginning with However, it is likely that these types of samples….” has been re-worded as “However, it is likely that these samples comprise a mixture of sensitive and resistant parasite populations, thus making it difficult to conclusively identify which parasite genotypes, and mutations in the Cyto b gene or TaPIN1 gene, are most strongly linked to buparvaquone resistance.” (Lines 91-94 in the modified manuscript).

Comment. Line 109: were

The Material and Methods section “parasite material” has been re-written to improve flow: “A total of 140 T. annulata schizont-infected cell lines obtained from animals in the acute phase of theileriosis and 168 T. annulata piroplasm-positive blood samples obtained from healthy carrier animals were used in the present study.” (Lines 105-107 in the modified manuscript).

Comment. Line 113 and other places: Better to write “is shown in supporting information (Fig S1). Correct other places also.

This sentence has been re-written as “A map illustrating the geographical location of the sampling sites is shown in supporting information (S1 Fig) with isolate details summarised in S1 Table.” (Lines 129-130 in the modified manuscript).

Comment. Line 126,127 and many other places: ml should be replaced by mL.

All references to “ml” in the text, figures and tables have been replaced by “mL”. Additionally, ‘μl’ has been replaced with ‘μL’.

Comment. Line 135: Revise writing “drug sensitive control parasite population”

The sentence “Theileria annulata / Ankara (D7), a clonal cell line known to be susceptible to buparvaquone, was used as a drug sensitive control parasite population for the MTT assay” was re-worded as follows: “Theileria annulata / Ankara (D7), a clonal cell line known to be susceptible to buparvaquone, was used as a drug-sensitive control parasite population”. (Lines 143-144 in the modified manuscript).

Comment. Line 165: Remove “available” which is repeating

The excess word “available” has been deleted. (Line 175 in the modified manuscript).

Comment. Line 168: Taq should be italicized

The word Taq has been written in italic. (Line 178 in the modified manuscript).

Comment; Line 147, 156: It is difficult for a naïve reader to understand what is A10/AT3. In the “parasite material” section you may add another paragraph to reveal how you have named the isolates.

A new paragraph has been added in the “parasite material”section explaning the meaning of the codes of the T. annulata isolates. 

“In order to obtain T. annulata schizont infected cell lines, 10 mLmL blood samples were collected into heparinised tubes from cattle showing clinical signs of the disease before and/or after buparvaquone treatment(s). In vitro establishment of schizont- infected cells from peripheral mononuclear cells (PBM) werewas carried out as previously described [23]. Once the cell line is established, a code was given to each isolate according to the site, where the blood samples were taken. The letters M, S, G, CN, AC, C, A, K, and N stand for the provinces of Center, Söke, Germencik, Kocarlı, İncirliova, Cine, Akçaova, Kösk and Nazilli, respectively. The numbers following these letters indicate the identification number of the animal from which the T. annulata schizont cell culture was established, viz. A10 stands for the 10th animal obtained from the Akçaova province. This was followed by the timing (BT: before or AT: after treatment) and the number of buparvaquone treatment. Accordingly, A10/BT stands for the isolate obtained from the 10th animal in Akçaova province before the buparvaquone treatment. Similarly, A21/AT4 stands for the isolate obtained from the 21st animal in Akçaova province after the 4th buparvaquone treatment.” (Lines 111-124 in the modified manuscript).

Comment. Line 192: sequencing need not be capitalized for the first letter

The first letter of sequencing has been written as lowercase. (Line 197 in the modified manuscript).

Comment. Line 204-206: This is part of the results. Rewrite

The sentence “Obtained sequence data revealed the presence of two mutations (V135A and P253S) at the putative drug binding domain of Cyto b gene in clonal cell lines A10 and A21, respectively, with respect to the published T. annulata genome sequence” has been deleted. The sentence following this “To determine the frequency of these mutations among the T. annulata isolates, clonal cell lines and blood samples from carrier animals which form this study, an allele-specific PCR (AS-PCR) was developed” has been re-worded as follows: “An allele-specific PCR (AS-PCR) was developed to determine the frequency of V135A and P253S mutations among the T. annulata isolates, clonal cell lines and blood samples from carrier animals used in the present study.” (Lines 217-219 in the modified manuscript).

Comment. Line 209-211: the primer design is not clear from the sentence written

The paragraph has been amended as “An allele-specific PCR (AS-PCR) was developed to determine the frequency of V135A and P253S mutations among the T. annulata isolates, clonal cell lines and blood samples from carrier animals used in the present study. When designing the primers, each single point mutation was positioned at the 3′ end of the forward primers, as previously indicated [28- 31]. Two forward (one wild-type and one mutant) and one reverse (wild-type) primers were designed for each mutation. The optimum annealing temperatures were adjusted by performing a gradient PCR for each reaction.” (Lines 217-223 in the modified manuscript).

Comment. Line 211: primers

The word “primer” has been corrected as “primers”. (Line 221 in the modified manuscript).

Comment. Line 246,298: SPSS/MTT: Do not begin a sentence with an abbreviation

Line 246: The sentence beginning with SPSS has been changed to “Statistical analyses were performed using SPSS software for Windows (Version 25.0. Armonk, NY: IBM Corp., USA).” (Lines 256-257 in the modified manuscript).

Line 254: The sentence beginning with MTT has been changed to “The results of the MTT analyses conducted to determine the susceptibility of infected cell lines to buparvaquone demonstrated that these T. annulata field isolates showed a variety of phenotypes (S1 Table).“ (Lines 268-270 in the modified manuscript).

Line 121: The title “MTT colourimetric assay” has been changed as “Tetrazolium dye (MTT) colourimetric assay” (Line 131 in the modified manuscript).

Line 298: The sentence “MTT analysis was performed with these clonal lines and IC50 values are given in (Table 1)” has been corrected as “Results of MTT analyses performed with these clonal lines and IC50 values are given in Table 1.” (Lines 309-310 in the modified manuscript).

Comment. Line 255: observed/calculated instead of obtained

The word “obtained” has not be changed in the manuscript. However, if the referee thought it would be better we could do it. (Line 270 in the modified manuscript).

Comment. Line 263-265: rewrite.

The sentence “When compared to before treatment a more dramatic (almost a tenfold) increase in the IC50 values of isolates from animals A10 that received repeated buparvaquone treatment was observed.” has been re-written as “Similarly, the IC50 values of isolates obtained from animal A10 before (A10/BT) and after a third buparvaquone treatment (A10/AT3) were 16.75 ng/mL and 135 ng/mL, respectively (Fig 2A, S1 Table). This demonstrates a dramatic (almost ten-fold) increase in IC50 values for infected cell lines isolated from animal A10 after repeated buparvaquone treatment.” (Lines 279-283 in the modified manuscript).

Comment. Line 270,275: Figure headings should be changed.

Line 270: The title “MTT test results of T. annulata field isolates” has been changed as “MTT assay of T. annulata infected cell lines.” (Line 285 in the modified manuscript).

Line 275: The title “MTT results of the isolates A10 (A) and A21 (B) after treatment with repeated buparvaquone administrations” has been changed as “MTT assay of T. annulata infected cell lines obtained from animals A10 (A) and A21 (B) following repeated buparvaquone treatments.” (Lines 290-291 in the modified manuscript).

Line 375: Although not specified by the reviewer 2, the title of Fig 5. has also been corrected. The title “Fig 5. MTT results of the clones obtained from A10/BT (A) and A21/AT1 (B) isolates” has been changed as “Fig 5. MTT assay of cloned cell lines from A10/BT (A) and A21/AT1 (B) isolates.” (Line 391-393 in the modified manuscript).

Comment. Line 284: Macroschizont is a parasite? The sentence may be rewritten. Also, whether the macro/microschizont concepts are still there?

The sentence “The genetic diversity of the parasites contained within the macroschizont-infected cell lines with high…” has been re-written as “The genetic diversity of the parasites within the T. annulata schizont-infected cell lines with high (A10/AT3 and A21/AT4), medium (A16/AT1, G3/BT) and low (A9/BT, N3/BT) IC50 values was assessed by micro-satellite analysis for five of the lines.” (Lines 295-297 in the modified manuscript).

We agree with the reviewer that we should not use the terms macro/micro-schizont as they are no longer in common use and the terms “schizont and merozoite” appear to be more appropriate. However, it seems that it has been a habit for many researchers working on Theileria spp. to describe the Koch bodies in leukocytes as is the case in many papers recently published on Theileria spp. Nevertheless, the term “macroschizont” has been changed with the term “schizont” throughout the manuscript. 

Comment. Line 292,372,375,390: Remove full stop after heading

According to the formatting guidelines of PlosONE, there should be a dot at the end of the title of a Table or a Figure, viz. “Fig 1. This is the Fig 1 Title. This is the Fig 1 legend.” and “Table 1. This is the Table 1 Title.” However, we stand-by to remove all the dots based on the journal’s advice.

Comment. Line 299: Remove parenthesis

The parenthesis has been removed. (Line 310 in the modified manuscript).

Comment ; Line 364: “obtained” may be changed

The word “obtained” has been removed. The sentence beginning with “The results obtained revealed that, compared to clones….” has been re-worded as “The results revealed that, compared to clones without the mutation (mean IC50:1.19 ng/mL; range: 0.73-1.51 ng/mL), clones with V135A were resistant even at very high concentrations (mean IC50: 45.28 ng/mL; range: 12.62-65.34 ng/mL) of buparvaquone (Table 4, Fig 5A).” (Lines 380-383 in the modified manuscript).

Comment. Line 415: “These results indicate that” may be replaced with Hence,

“These results indicate that” has been replaced by “Hence” (Line 425 in the modified manuscript).

Comment. Line 427-429: Rewrite the sentence

The sentence “Whether the emergence of the two resistant parasite populations identified in the present study is associated with an increase of resistance in the filed via tick transmission remains to be determined” has been re-written as “Further studies are warranted to investigate the rate at which drug resistance disseminates in T. annulata populations, taking account of vector transmission intensity and drug pressure.” (Lines 440-442 in the modified manuscript).

Comment. Line 441: demonstrated

This paragraph has been deleted while reducing the discussion of the manuscript.

Comment. Line 452- 463, 491-512: The information may be tabulated.

The information between lines 452-463 was tabulated as supporting information S4 Table (Lines 451-454 in the modified manuscript) and the information between lines 491-512 was already tabulated as supporting information S2 Table (Lines 491-493 in the modified manuscript).

 

ADDITIONAL CHANGES MADE IN THE MANUSCRIPT. 

Title page:

 Line 9: The order of authors has been changed as Selin Hacılarlıoglu, Huseyin Bilgin Bilgic, Serkan Bakırcı, Andrew Tait, William Weir, Brian Shiels, Tulin Karagenc (Lines 8-9 in the modified manuscript).

Affiliation of the authors has been changed as below:

Line 14: The word “Turkiye” has been chanced as “Türkiye” (Line 14 in the modified manuscript).

Line 16: “School of Veterinary Medicine” has been corrected as “School of Biodiversity, One Health and Veterinary Medicine” (Line 16 in the modified manuscript).

Line 20-22 “3 Institute of Biodiversity, Animal Health and Comparative Medicine, College of Medicine, Veterinary and Life Sciences, University of Glasgow, Bearsden Road, Glasgow, G61 1QH, UK” has been deleted.

Abstract: 

Line 33-35: The sentence “However, an increase in the rate of treatment failures has been observed in recent years. This raised the possibility of resistance to buparvaquone, proposed to be associated with a mutation(s) in the cytochrome b gene (Cyto b) of T. annulata” has been re-worded as “However, an increase in the rate of buparvaquone treatment failures has been observed in recent years, raising the possibility that resistance to this drug is associated with the selection of T. annulata genotypes bearing mutation(s) in the cytochrome b gene (Cyto b)” (Lines 29-32 in the modified manuscript).

Line 36-38: The sentence “The aims of the present study were two-fold: (1) to demonstrate if there is an association between mutations in the T. annulata Cyto b gene and resistance to buparvaquone, and (2) to determine the frequency of these mutations in infected cattle in the Aydın region of Turkey” has been re-worded as “The aim of the present study was: (1) to demonstrate whether there is an association between mutations in the T. annulata Cyto b gene and selection of parasite-infected cells resistant to buparvaquone and (2) to determine the frequency of these mutations in parasites derived from infected cattle in the Aydın region of Türkiye.” (Line 32-35 in the modified manuscript).

Line 39: The word “examined” has been replaced by “assessed” and “infected cells” has been changed as “schizont-infected cells” (Line 36 in the modified manuscript).

Line 42: The word “calorimetric” has been corrected as “colourimetric” (Line 39 in the modified manuscript).

Line 42-43: The sentence “The DNA sequence of Cyto b gene of cell lines identified as resistant or susceptible were determined” has been re-worded as “The DNA sequence of the parasite Cyto b gene from cell lines identified as resistant or susceptible was determined.” (Lines 39-40 in the modified manuscript).

Line 44: The word “detected” has been replaced by “identified” (Line 41 in the modified manuscript).

Line 44-45: The sentence “Of these, two nonsynonymous mutations (V135A and P253S) were located at putative buparvaquone binding regions of the Cyto b gene.” has been re-worded as “Two of the nonsynonymous mutations resulted in the substitutions V135A and P253S which are located at the putative buparvaquone binding regions of cytochrome b.” (Lines 41-43 in the modified manuscript).

Line 45-48: “Allele-specific PCR (AS-PCR) analyses demonstrated the presence of the mutations V135A and P253S with a frequency of 3.90 % and 3.57 % respectively and further revealed an increase in the frequency of both mutations over the years” has been re-worded as “Allele-specific PCR (AS-PCR) analyses detected the V135A and P253S mutations at a frequency of 3.90 % and 3.57 % respectively in a regional study population and revealed an increase in the frequency of both mutations over the years.” (Lines 43-45 in the modified manuscript).

Line 48-49: The sentence beginning with “A53P mutation of TaPIN1,….” has been re-worded as “The A53P mutation of TaPIN1 of T. annulata, previously suggested as being involved in buparvaquone resistance, was not detected in any of the clonal cell lines examined in the present study.” (Lines 45-47 in the modified manuscript). 

Line 49-52: “These data strongly suggest that V135A and P253S mutations detected at the putative binding sites of buparvaquone in the Cyto b gene play a significant role in conferring resistance of T. annulata to buparvaquone, whereas the role of mutations in TaPIN1 was found to be more equivocal” has been re-written “The obtained data strongly suggest that the genetic mutations resulting in V135A and P253S detected at the putative binding sites of buparvaquone in cytochrome b play a significant role in conferring, and promoting selection of, T. annulata genotypes resistant to buparvaquone, whereas the role of mutations in TaPIN1 is more equivocal” (Line 47-51 in the modified manuscript).

Introduction:

Line 60: The word “causes” has been replaced by “has” (Line 59 in the modified manuscript).

Line 62-63: “which provides solid immunity against homologous challenge and partial protection against heterologous strains” has been deleted.

Line 64: “is questionable” has been added after “endemic regions” (Line 62 in the modified manuscript). 

Line 65: The word “remain” has been replaced by “remains” and “The third applied method” has been replaced by “The third, most widely applied method” (Line 63 in the modified manuscript).

Line 66: The word “Currently” has been removed (Line 65 in the modified manuscript).

Line 68: The sentence “in countries where the disease is endemic” has been deleted in the modified manuscript.

Line 69: “…the rate of cases of treatment failure of tropical theileriosis has…” has been re-worded as “….the rate of buparvaquone treatment failure has been reported in…” (Line 67 in the modified manuscript).

Line 72: The word “unclear” has been replaced by “not fully known” (Line 69 in the modified manuscript).

Line 74: “Evidence gathered from” has been deleted in the modified manuscript..

Line 77 the word “parasite” has been deleted in the modified manuscript..

Line 77-80 The sentence beginning with “Therefore, it was logical to propose that a mutation in Cyto b of T. annulata, especially……..” has been re-worded as “Based on the structural similarities between buparvaquone and atovaquone, it was logical to propose that a mutation in Cyto b of T. annulata, particularly at the binding site of ubiquinone, would be associated with resistance to buparvaquone. [8- 11]” (Lines 75-78 in the modified manuscript).

Line 81: The word “known” has been replaced by “thought” (Line 79 in the modified manuscript).

Line 83: The word “PIN1” has been deleted in the modified manuscript. 

Line 84-87: “It was also demonstrated that TaPIN1-induced transformation process is inhibited by buparvaquone and that, in the presence of a mutation (A53P) at residue 53 at the catalytic loop of TaPIN1, buparvaquone failed to inhibit PIN1 activity [22]” has been re-worded as “The TaPIN1-induced transformation process is inhibited by buparvaquone while a mutation (A53P) at residue 53 of the catalytic loop of TaPIN1 reverses the ability of buparvaquone to inhibit PIN1 activity [22]” (Lines 82-84 in the modified manuscript).

Line 87-88: The sentence” The A53P mutation was identified in drug-resistant isolates from both Tunisia and Sudan [12, 22]” has been re-worded as “In further studies, the A53P mutation was identified in some, but not all, buparvaquone-resistant isolates from both Tunisia and Sudan [12, 22]” (Lines 84-85 in the modified manuscript).

Line 88-89 The sentence “It is suggested on the basis of these observations that A53P mutation in TaPIN1 might also be involved in resistance to buparvaquone [12, 22]” has been re-worded “On the basis of these observations it was proposed that the A53P mutation in TaPIN1 could be involved in resistance to buparvaquone in field populations of the parasite [12, 22], but the strength of this association requires further validation” (Lines 86-88 in the modified manuscript).

Line 90: “The majority of the studies to date investigating…” has been re-worded as “The majority of previous studies investigating…” (Line 89 in the modified manuscript).

Line 92-94: “However, it is likely that these types of samples comprise a mixture of sensitive along with resistant parasite populations, thus making it difficult to determine which parasite particular genotypes may be resistant to buparvaquone” has been re-worded as “However, it is likely that these samples comprise a mixture of sensitive and resistant parasite populations, thus making it difficult to conclusively identify which parasite genotypes, and mutations in the Cyto b gene or TaPIN1 gene, are most strongly linked to buparvaquone resistance” (Line 91-94 in the modified manuscript).

Line 94: The word “Therefore” has been deleted in the modified manuscript.

Line 99-101: The sentence “T. annulata Cyto b gene was then sequenced to characterize putative buparvaquone resistance-associated mutations exist” has been re-worded as “T. annulata Cyto b and TaPIN1 genes were sequenced to determine whether putative buparvaquone resistance-associated mutations were present” (Line 99-100 in the modified manuscript).

Line 103: The article “an” added before allele-specific PCR. The word “in order” has been deleted and the word “measure” has been replaced by “investigate”. The word “drug” has been added before “resistance” (Lines 101-102 in the modified manuscript). 

Line 104-105 The sentence “Sequence analyses of TaPIN1 gene were also performed to examine if there is any association of A53P mutation with resistance of the infected lines to buparvaquone” has been deleted in the modified manuscript.

Material methods: 

Line 122-124 “The sentence beginning with “Susceptibility of T. annulata cell line isolates (S1 Table) to buparvaquone ……..” has been re-worded as “Susceptibility of T. annulata cell line isolates (S1 Table) to buparvaquone was evaluated by using an MTT colourimetric assay to determine the proliferative index of infected culture under various doses of buparvaquone, as described [24] but with slight modification” (Lines 132-134 in the modified manuscript).

Line 128-130 The sentence “Buparvaquone (ALKE Healthcare Products Inc., Turkiye) was kept in powder form at 4 ºC and a solution prepared just prior to use, as described previously [25]” has been re-worded as “Stock buparvaquone (ALKE Healthcare Products Inc., Türkiye) solution (at 4000ng/mL) was prepared just prior to use, as previously described [25]” (Lines 138-139 in the modified manuscript).

Line 135-136 The sentence “All cultivated test and control cell lines were maintained under experimental conditions at 37 ºC and 5 % CO2 for three days” has been re-worded as “All cultivated cell lines were maintained at 37 ºC and 5 % CO2 for three days” (Lines 145-146 in the modified manuscript).

Line 142 The word “on” has been chanced by “using” and “the” has been deleted (Line 151 in the modified manuscript).

Line 156 “Following genotyping” at the beginning of the sentence has deleted in the modified manuscript.

Line 160 “For the drug” has been added at the end of the sentence (Line 170 in the modified manuscript).

Line 160-164: The sentence beginning with “DNA was extracted for two clones with the highest IC50 value (Table 1)……” has been re-worded as “DNA was extracted from two clones with the highest IC50 value (Table 1) for each cell line. The DNA was then used in three independent PCRs to amplify a 1,089 bp region of the Cyto b gene of T. annulata. Primers used were: forward (5'-ATG AAT TTG TTT AAC TCA CAT TTG C- 3') and reverse (5'- TGC ACG AAC TCT TGC AGA GTC- 3')” (Lines 170-171 in the modified manuscript).

Line 164-166: The sentence “Primers were designed to specifically amplify the open reading frame (ORF) of this gene on the basis of available publicly available sequence data (GenBank accession no: XM949625)” has been re-worded as “These were designed to specifically amplify the open reading frame (ORF) of the gene based on publicly available sequence data (GenBank accession no: XM949625)” (Lines 171-174 in the modified manuscript).

Line 166-169: The sentence beginning with “PCR conditions comprised 45 mM Tris–HCl……” has been re-worded as “PCR was performed with 45 mM Tris–HCl (pH 8.8), 11 mM (NH4)2SO4, 4.5 mM MgCl2, 0.113 mg/mL BSA, 4.4 µM EDTA, 1 nM dNTPs, 10 µM of each primer, 1 U Taq DNA polymerase (Solis BioDyne, Estonia) and 40 ng template DNA in a total volume of 50 µl” (Line 176-178 in the modified manuscript).

Line 169: “PCR was carried” out has been deleted from the beginning of the sentence.

Line 174: Table 1; “following treatment” has been chanced by “treated” (Line 184 in the modified manuscript).

Line 175: The word “which” has been added after “cell lines” (Line 185 in the modified manuscript).

 Line 176: The footnote of the table has been changed as “(*) indicates clonal cell lines which were used for sequence analysis. (a, b) Different lower case superscript letters indicate statistical significance (P < 0.05). IC50 values obtained from the clones of the cell lines A10/AT3 and A21/AT4 was significantly different from IC50 values of the clones of the cell lines A9/BT, A16/AT1, G3/BT and N3/BT. There was no significant difference in IC50 values between A10/AT3 and A21/AT4, and among the A9/BT, A16/AT1, G3/BT and N3/BT (P > 0.05)” (Line 187-189 in the modified manuscript).

Line 178-185: The paragraph beginning with “In addition to the Cyto b gene ……” has been re-worded as “The TaPIN1 gene was screened for the presence of the mutation (A53P) at amino acid residue 53, within the catalytic loop of the predicted protein. For this purpose, four resistant clonal cell lines (A10/AT3/cl8 and A10/AT3/cl9 with the V135A mutation in Cyto b and A21/AT4/cl3 and A21/AT4/cl8 with the Cyto b P253S mutation) and three buparvaquone sensitive (A9/BT/cl5, G3/BT/cl2 and N3/BT/cl4) clonal cell lines were used. PCR amplification of a 900 bp region of the TaPIN1 gene was performed as described by Marsolier et al. [22] using the following primers, Forward: 5' GTC TGT CAA ATA GGT AGA AAT C- 3' and Reverse: 5' GAG AGG AAG TTG AAT CAA ACA T- 3', as detailed above except that an annealing temperature of 56.6 ºC was used. PCR amplicons of TaPIN1 were cloned as described for the Cyto b gene.” and moved to the lines between 200-209 in the modified manuscript. 

Line 187: “within putative QO1 and QO2 domains of the protein” has been re-worded as “corresponding with putative QO1 and QO2 domains of the encoded protein” (Line 192 in the modified manuscript).

Line 191: “(InvitrogenTM)” has been chanced by “(Invitrogen, Germany)” (Line 196 in the modified manuscript).

Line 201: “and” has been added before “MK693135 (A16/AT1/cl10)” (Line 215 in the modified manuscript).

Line 214: The word “reveal” has been replaced by “detect” and “the” has been deleted (Line 223 in the modified manuscript).

Line 224-225: The sentence “2 μl of template DNA was used for each PCR reaction” has been re-worded as “and 40 ng of template DNA” and has been moved at the end of the sentence (Line 233 in the modified manuscript).

Line 228: “P253S mutation and 72 ºC for 1 min 20s with a final extension of 72 ºC for 10 min” has been re-worded as “P253S mutation annealing step and 72 ºC for 1 min 20s extension, with a final extension of 72 ºC for 10 min” (Line 237 in the modified manuscript).

Line 230: The preposition “of” has been replaced by “from” (Line 240 in the modified manuscript).

Line 230-232: The sentence beginning with “The DNA samples of clonal cell lines comprising ….” has been re-worded as “DNA samples from cloned lines possessing either the V135A (R1) or P253S (R2) mutations were mixed with DNA from a wild-type sensitive clonal cell line (S) (R1/S or R2/S)” (Lines 240-242 ------------- in the modified manuscript).

Line 232-234: The sentence beginning with “For the specificity of the AS-PCR, the DNA samples….” has been re-worded as “To test the specificity of the AS-PCR, DNA samples obtained from lines with V135A (R1) or P253S (R2) mutations were diluted (0.2 - 20 ng/µl) with each other (R1/R2: 1/10, 1/20; 1/50, 1/100 or R2/R1: 1/10, 1/20; 1/50, 1/100)” (Lines 242- 244 in the modified manuscript).

Line 235: The article “The” has been deleted from the beginning of the sentence (Line 244 in the modified manuscript).

Line 236-238: The sentence “EDTA blood samples collected from carrier animals were used to extract DNA using the Promega Wizard genomic DNA extraction kit (Madison, WI, USA) following the manufacturer’s instructions” has been re-worded as “DNA was extracted from EDTA blood samples collected from carrier animals using the Promega Wizard genomic DNA extraction kit (Madison, WI, USA), following the manufacturer’s instructions” (Lines 246-248 in the modified manuscript). 

Line 238-240: The sentence “Extracted DNA was resuspended in 100 μl rehydration buffer and stored at −20 ºC until used. All DNA samples were screened for the presence of mutations using the AS-PCR protocol described above” has been re-worded as “Extracted DNA was resuspended in 100 μL rehydration buffer and stored at −20 ºC. The DNA samples were then screened for Cyto b mutations using the AS-PCR protocol” (Lines 248-250 in the modified manuscript).

Line 242: Article “the” has been added before Shapiro-Wilk test (Line 252 in the modified manuscript).

Line 243: “were” has been chanced by “was” (Line 253 in the modified manuscript).

Line 244: The word “analyzed” has been corrected by “analysed” and article “the” has been added before Kruskal–Wallis. Articles “the” and “a” has been added after Kruskal–Wallis and before Bonferroni respectively (Lines 254-255 in the modified manuscript).

Results:

Line 255-257: The sentence beginning with “Based on obtained IC50 values, the isolates were divided into three groups with low (1-3 ng/ml), moderate (3-7 ng/ml) and high (>7 ng/ml) IC50 values” has been re-worded as “Based on obtained IC50 values, the isolates were divided into three groups representing low (1-3 ng/mL), moderate (3-7 ng/mL) and high (>7 ng/mL) resistance to buparvaquone” (Line 270-272 in the modified manuscript).

Line 257: The word “While” has been moved before the medium and high” . (Line 272 in the modified manuscript).

Line 258: The preposition “of” has been deleted in the modified manuscript.

Line 259-261: The sentence beginning with “When compared to the first treatment…….” has been re-written as “Evidence for selection of resistant parasites was found as, compared to the first drug treatment, a marked (five-fold) increase in the IC50 values of isolates from animal A21 that received repeated buparvaquone treatment was observed over time” (Lines 274-277 in the modified manuscript).

Line 265-268: The sentence beginning with “The IC50 values of three isolates obtained from….” has been re-written as “Thus, the IC50 value of the isolates obtained from animal A21 after the first (A21/AT1) compared to the fourth (A21/AT4) buparvaquone injections were 16.75 ng/mL and 73.79 ng/mL, respectively (Fig 2B, S1 Table)” and moved to between lines 277-279 in modified manuscript.

Line 283: The title “Genotype of isolates” has been re-written as “Evidence of genotypic selection in isolates after drug treatment” (Line 294 in the modified manuscript).

Line 284: The sentence beginning with “The genetic diversity of the parasites within……” has been re-worded as “The genetic diversity of the parasites within the T. annulata schizont-infected cell lines with high (A10/AT3 and A21/AT4), medium (A16/AT1, G3/BT) and low (A9/BT, N3/BT) IC50 values was assessed by micro-satellite analysis for six of the lines” (Line ------------- in the modified manuscript).

Line 286: The word “isolates” has been replaced by “cell lines” (Lines 295-297 in the modified manuscript).

Line 287-290: The sentence beginning with “Repeated-buparvaquone treatment……” has been re-worded as “Repeated-buparvaquone treatment reduced the diversity of the parasite population in lines isolated from animals A10 and A21. Only a single parasite genotype was detected following the third buparvaquone treatment in the cell line of the isolate A10/AT3 isolate (Table 2)” (Line 298-301 in the modified manuscript).

Line 291: Table 2 has been changed. The results of micro-satellite analysis were included for all the cell lines tested in Table 2. (Lines 303-304 in the modified manuscript).

Line 297-298: The sentence beginning with “The six field isolates were cloned by limiting ……” has been re-written as “Six cell lines from field isolates corresponding to the three different levels buparvaquone resistance were cloned by limiting dilution and for each isolate, ten bona fide clones were selected. The presence of a single genotype in each clonal line was confirmed by multilocus genotyping” (Line 306-309 in the modified manuscript).

Line 298: The sentence beginning with “MTT analysis…” has been re-worded as “Results of MTT analyses performed with these clonal lines and IC50 values are given in Table 1” (Lines 309-310 in the modified manuscript).

Line 300: The preposition “in” has been chanced by “for” (Line 310 in the modified manuscript).

Line 303: The sentence “Taken together the results indicate that treatment with buparvaquone has selected parasite genotypes that may harbour mutations conferring resistance to the drug” has been added at the end of paragraph (Line 314-315 in the modified manuscript).

Line 305-319: The paragraph beginning with “The complete Cyto b gene (1,089 bp) of two….” has been re-worded as “The Cyto b gene (1,089 bp) was sequenced for two representative clonal cell lines derived from each of the six isolates with high (A10/AT3 and A21/AT4), medium (A16/AT1, G3/BT) and low (A9/BT, N3/BT) IC50 values. This was done in order to screen for the presence of non-synonymous mutations at the ubiquinone binding site, hypothesised to be associated with parasite resistance to buparvaquone. Cyto b sequences from the clonal cell lines were then aligned to and compared with the Cyto b gene (XM949625) sequence of the reference T. annulata (C9) genome. Six positions with non-synonymous mutations and six positions with synonymous nucleotide mutations were detected (Table 3). The six non-synonymous mutations sites were located at positions 151, 404, 436, 679, 757 and 1015 of the Cyto b gene, predicted to result in the following amino acid substitutions: M51L, V135A, A146T, V227M, P253S and A339V, respectively. Two of these non-synonymous mutations, between codons 116-144 (Qo1) and 242-286 (Qo2), respectively generate amino acid changes V135A and P253S located in the putative ubiquinone-binding sites (QO) [32, 33]. Substitution V135A at the Qo1 binding site was predicted in clones of A10/AT3, whereas substitution P253S at the Qo2 binding site was predicted for clones of A21/AT4 (Fig 3)” (Lines 317-331 in the modified manuscript). 

Line 320-324: The title and the legend of the Fig 3 has been changed as “Alignments of putative Qo1 and Qo2 domains of Cyto b. Predicted amino acid sequences of Qo1 and Qo2 domains encoded by the Cyto b gene from the published T. annulata / C9 genome sequence and different T. annulata isolates with treatment failure history. Putative Qo1 and Qo2 domains are located between 116–144 and 238–273 amino acids of the protein around the ubiquinone binding site. Predicted amino acid substitutions associated with buparvaquone resistance are marked with ◊” (Lines 332-337 in the modified manuscript).

Line 326: The footnote of the Table 3 “(*) indicates nonsynonymous mutation detected in related isolate” has been re-worded as “(*) indicates the non-synonymous mutations sites” (Line 340 in the modified manuscript).

Line 328-336: The paragraph beginning with “Sequence analyses of catalytic loop of TaPIN1 gene………” has been re-worded as “Sequence analyses of the TaPIN1 gene of the same twelve resistant and susceptible clonal cell lines demonstrated that the A53P substitution in the catalytic loop of the predicted protein reported by Marsolier et al. [22] was not present in any of the buparvaquone resistant and susceptible clonal cell lines examined (Fig 4, S2 Table). However, several other mutations in TaPIN1 were detected which were predicted to result in amino acid substitution, including I2F, T22I, T22A, I23N, A26P, A26T, L78P and R96K. While the majority of the mutations were observed in sensitive clonal cell lines, three non-synonymous mutations (I2F, T22I and A26T) were detected only in the resistant clonal cell lines from A21/AT4 (clone 3 and clone 8). Additional mutations resulting in substitutions at amino acid positions 22 (T22A) and 26 (A26P) were also detected among the susceptible clonal cell lines.” (Lines 342-351 in the modified manuscript).

Line 338-342: The title and the legend of the Fig 4 has been changed as “Alignments of TaPIN1 nucleotide (A) and predicted amino acid (B) sequences. Sequences derived from a buparvaquone resistant Tunisian isolate and various Turkish T. annulata isolates with treatment failure history were compared for mutations with the T. annulata / C9 reference genome, obtained from PiroplasmDB.org database; the systematic identifier for TaPIN1 is TA18945. Nonsynonymous substitutions detected among alignments were indicated with an asterisk (*) coloured in yellow” (Lines 353-358 in the modified manuscript).

Line 344-345 The title “The frequency of mutations V135A and P253S in T. annulata clonal cell lines” has been re-worded as “Development of AS-PCR to measure frequency of Cyto b V135A and P253S mutations in drug resistant T. annulata cell lines” (Lines 360-361 in the modified manuscript).

Line 346-348: The sentence “The frequency of V135A and P253S mutations detected at the putative drug binding domain of the Cyto b gene in resistant clonal cell lines (A10 and A21) were determined using an AS-PCR assay” has been re-written “The frequency of V135A and P253S substitutions detected at the putative drug binding domain of Cytochrome b among resistant clonal cell lines (A10 and A21) was determined using an AS-PCR assay”. (Lines 362-364 in the modified manuscript).

Line 349-369: The paragraph beginning with “AS-PCR assays using diluted (0.2 - 20 ng/µl) DNA ………” has been re-worded as “The developed AS-PCR assay was validated using diluted (0.2 - 20 ng/µl) DNA samples from lines with the V135A or P253S substitutions, demonstrated that 400 pg/µl DNA was sufficient to detect the targeted nucleotide mutations. Based on these data, clones of A9/BT (n = 45), A10/AT3 (n = 45), A16/AT1 (n = 40), A21/AT4 (n = 48), G3/BT (n = 9) and N3/BT (n = 42) isolates were analysed with the AS-PCR (S3 Table) to determine the presence of V135A and P253S mutations (S2 Fig). The mutation conferring V135A was detected in all 45 clones of isolate A10/AT3, while the mutation generating P253S was detected in all 48 clones of isolate A21/AT4. Following this result, corresponding pre-treatment (A10/BT) and early treatment (A21/AT1) isolates were cloned and sequenced to determine whether wild-type alleles were originally present and whether selection of resistant genotypes had occurred. AS-PCR analyses of clonal cell lines derived from these isolates revealed that while the mutation conferring V135A is present in the majority (38/41) of the clonal cell lines obtained from the isolate A10/BT, the mutation resulting in P253S is present only in 8 of 49 clonal cell lines obtained from isolate A21/AT1. For phenotypic characterisation, six clonal cell lines from isolates A10/BT and A21/AT1 (three with a mutation and three without any of the resistance associated mutations) were examined with the MTT assay. The results revealed that, compared to clones without the mutation (mean IC50:1.19 ng/mL; range: 0.73-1.51 ng/mL), clones with V135A were resistant even at very high concentrations (mean IC50: 45.28 ng/mL; range: 12.62-65.34 ng/mL) of buparvaquone (Table 4, Fig 5A). Similarly, the clones derived from isolate A21 with the P253S substitution had higher IC50 values (mean IC50: 21.0 ng/mL; range: 6.36-37.70 ng/mL) than clones without the associated nucleotide mutation (mean IC50: 2.23 ng/mL; range: 1.97-3.17; Table 4, Fig 5B).” (Lines 365-386 in the modified manuscript).

.

Line 371-372: The title of Table 4 has been re-written as “IC50 values of A10/BT and A21/AT1 clones with substitutions V135A and P253S” (Line 388 in the modified manuscript).

Line 379: The title “The frequency of V135A and P253S mutations in the field” has been re-worded as “Frequency of V135A and P253S mutations in the field isolates” (Line 295 in the modified manuscript).

Line 381-389: The paragraph beginning with “Having determined such a strong ……..” has been re-worded as ‘Having demonstrated a strong association between buparvaquone resistance and the presence of mutations in Cyto b that result in V135A and P253S substitutions, the AS-PCR developed for each mutation was used to measure the emergence and spread of drug resistance in the field. A total of 168 randomly selected T. annulata-positive blood samples and 140 T. annulata cell lines were tested. The AS-PCR results revealed that the V135A mutation was present in 10 of 168 carrier cattle (5.95 %) and 2 of 140 (1.42%) isolates, whereas the P253S mutation was identified in 1 (0.59 %) carrier animal and 10 (7.14 %) isolates (Table 5). The frequency of mutations for V135A and P253S across all samples was 3.89% and 3.57%, respectively. It was apparent from the blood and cell line isolates examined in the two time periods, (A) 1998-2007 compared to (B) 2010-2011, that the frequency of the both mutations increased significantly in the latter period: V135A (P = 0.007), P253S (P = 0.000) (Table 5).” (Lines 396-406 in the modified manuscript). 

Line 391: A footnote has been added to the Table 5 “(*) denotes significant difference in mutated allele frequency in period A compared to period B” (Line ------------- in the modified manuscript).

Discussion:

The discussion has been substantially reduced and repetitive sections removed as indicated above. 

References:

While some of references were deleted from the reference list , one referense was added. The numbers of references have been also changed in the text. 

Deleted references

Vaidya AB, Mather MW. Atovaquone resistance in malaria parasites. Drug Resist Updat. 2000;3(5): 283-287. https://doi.org/10.1054/drup.2000.0157. PMID: 11498396.

McFadden DC, Tomavo S, Berry EA, Boothroyd JC. Characterization of cytochrome b from Toxoplasma gondii and Q(o) domain mutations as a mechanism of atovaquone-resistance. Mol Biochem Parasitol. 2000; 108(1):1-12. https://doi.org/10.1016/s0166-6851(00)00184-5. PMID: 10802314.

Erkut HM. Ege Bölgesinde sığırlarında piroplasmosis durumu ve tedavide yeni ilaçlamalar. Bornova Vet Aras Enst Derg. 1967; 8(16):120-130. Turkish.

Croft SL, Hogg J, Gutteridge WE, Hudson AT, Randall AW. The activity of hydroxynaphthoquinones against Leishmania donovani. J Antimicrob Chemother. 1992;30(6): 827-832. https://doi.org/10.1093/jac/30.6.827. PMID: 1289357.

Hyde JE. Drug-resistant malaria - an insight. The FEBS Journal. 2007; 274(18):4688-4698. https://doi.org/10.1111/j.1742-4658.2007.05999.x. PMID: 17824955.

Stokes BH, Ward KE, Fidock DA. Evidence of artemisinin-resistant malaria in Africa. N Engl J Med. 2022;386(14): 1385-1386. https://doi.org/10.1056/NEJMc2117480. PMID: 35388682.

Tepper O, Peled I, Fastman Y, Heinberg A, Mitesser V, Dzikowski R, et al. FIT-PNAs as RNA-sensing probes for drug-resistant Plasmodium falciparum. ACS Sens. 2022;7(1): 50-59. https://doi.org/10.1021/acssensors.1c01481. PMID: 34985283.

Stenhouse SA, Plernsub S, Yanola J, Lumjuan N, Dantrakool A, Choochote W, et al. Detection of the V1016G mutation in the voltage-gated sodium channel gene of Aedes aegypti (Diptera: Culicidae) by allele-specific PCR assay, and its distribution and effect on deltamethrin resistance in Thailand. Parasit Vectors. 2013;6(1): 253. https://doi.org/10.1186/1756-3305-6-253. PMID: 24059267.

Tan WL, Wang ZM, Li CX, Chu HL, Xu Y, Dong YD, et al. First report on co-occurrence knockdown resistance mutations and susceptibility to beta-cypermethrin in Anopheles sinensis from Jiangsu Province, China. PLoS ONE. 2012;7(1): e29242. https://doi.org/10.1371/journal.pone.0029242. PMID: 22272229.

Added reference

Bell AS, Huijben S, Paaijmans KP, Sim DG, Chan BH, Nelson WA, et al. Enhanced transmission of drug-resistant parasites to mosquitoes following drug treatment in rodent malaria. PLoS ONE. 2012;7(6): e37172. https://doi.org/10.1371/journal.pone. 0037172. PMID: 22701563.

---

## [Decision Letter · Decision Letter 1]

23 Nov 2022

PONE-D-22-20200R1Selection of genotypes harbouring mutations in the cytochrome b gene of Theileria annulata is associated with resistance to buparvaquonePLOS ONE

Dear Dr. Karagenc,

Thank you for submitting your manuscript to PLOS ONE. After careful consideration, we feel that it has merit but does not fully meet PLOS ONE’s publication criteria as it currently stands. Therefore, we invite you to submit a revised version of the manuscript that addresses the points raised during the review process.

We look forward to receiving your revised manuscript.

Kind regards,

Vikrant Sudan, PhD

Academic Editor

PLOS ONE

Journal Requirements:

Additional Editor Comments:

Minor revisions are requested by one of the reviewer. The changes need to be incorporated befor final decision.

Reviewers' comments:

Reviewer's Responses to Questions

**Comments to the Author**

1. If the authors have adequately addressed your comments raised in a previous round of review and you feel that this manuscript is now acceptable for publication, you may indicate that here to bypass the “Comments to the Author” section, enter your conflict of interest statement in the “Confidential to Editor” section, and submit your "Accept" recommendation.

Reviewer #1: All comments have been addressed

Reviewer #2: All comments have been addressed

2. Is the manuscript technically sound, and do the data support the conclusions?

Reviewer #1: Yes

Reviewer #2: Yes

3. Has the statistical analysis been performed appropriately and rigorously? 

Reviewer #1: Yes

Reviewer #2: Yes

4. Have the authors made all data underlying the findings in their manuscript fully available?

Reviewer #1: Yes

Reviewer #2: Yes

5. Is the manuscript presented in an intelligible fashion and written in standard English?

Reviewer #1: Yes

Reviewer #2: Yes

6. Review Comments to the Author

Reviewer #1: The manuscript is relevant in the era of emergence of drug resistance. The manuscript was thoroughly revised as per the requirements and I recommend for acceptance of the manuscript for publication

Reviewer #2: Comments on “Selection of genotypes harbouring mutations in the cytochrome b gene of Theileria annulata is associated with resistance to buparvaquone” PONE-D-22-20200R1

Minor changes suggested

Line 47-48: “The obtained data strongly suggest that the genetic mutations resulting in V135A and P253S detected at the putative binding sites of buparvaquone in cytochrome b play a significant role in conferring,” may be written as “The observed data strongly suggested -----------”

Line 79: “disrupting the ubiquinone synthesis pathway”????

Line147,149,239,243,365: μl maybe written as μL

Line 281: demonstrated

Line 307: of buparvaquone resistance

7. PLOS authors have the option to publish the peer review history of their article (what does this mean?). If published, this will include your full peer review and any attached files.

Reviewer #1: **Yes: **Siju Susan Jacob

Reviewer #2: **Yes: **REGHU RAVINDRAN

---

## [Author Response · Author response to Decision Letter 1]

14 Dec 2022

Dear Dr. Vikrant Sudan, 

Thank you for your response to our manuscript PONE-D-22-20200R1 entitled " Selection of genotypes harbouring mutations in the cytochrome b gene of Theileria annulata is associated with resistance to buparvaquone". We would also like to thank both the reviewers for their comments to improve the quality of the manuscript. 

The figures of the manuscript were uploaded to the PACE before resubmitting the manuscript and the corrected version of the figures by PACE were submitted to PLOS ONE. On the following pages, we present our point-by-point responses to the comments and suggestions made by the Academic Editor and reviewers. 

A revised version of our manuscript is also provided, where all modifications are shown in a separate file labeled 'Revised Manuscript with Track Changes’. Please be informed that all the changes made in the text as a response to reviewers are recorded using Track Changes in MS Word and highlighted in yellow collor.

Yours sincerely,

Tulin KARAGENC (DVM, PhD)

 

Response to Journal Requirements:

Comment: Please review your reference list to ensure that it is complete and correct. If you have cited papers that have been retracted, please include the rationale for doing so in the manuscript text, or remove these references and replace them with relevant current references. Any changes to the reference list should be mentioned in the rebuttal letter that accompanies your revised manuscript. If you need to cite a retracted article, indicate the article’s retracted status in the References list and also include a citation and full reference for the retraction notice.

The reference list has been checked and corrected according to the journal style. It is complete and we did not cite any papers that have been retracted.

Response to Reviewers' comments: 

Response to Reviewer 2:

Minor points

Comment. Line 47-48: “The obtained data strongly suggest that the genetic mutations resulting in V135A and P253S detected at the putative binding sites of buparvaquone in cytochrome b play a significant role in conferring,” may be written as “The observed data strongly suggested -----------”

Beginning of the sentence has been re-written as follows: “The observed data strongly suggested -------” (Line 48 in the modified manuscript).

Comment. Line 79: “disrupting the ubiquinone synthesis pathway”????

The sentence has been re-written as “It is also thought that buparvaquone acts as a peptidyl-prolyl isomerase PIN1 inhibitor.” (Line 79 in the modified manuscript)

Comment. Line147, 149, 239, 243, 365: μl maybe written as μL

All “μl” has been replaced by ‘μL’ in the text. 

Comment. Line 281: demonstrated

The word “demonstrates” has been replaced by “demonstrated”. (Line 281 in the modified manuscript)

Comment. Line 307: of buparvaquone resistance

The preposition “of” has been added before the “buparvaquone resistance” (Lines 306-307 in the modified manuscript).

---

## [Editor Report · Decision Letter 2]

19 Dec 2022

Selection of genotypes harbouring mutations in the cytochrome b gene of Theileria annulata is associated with resistance to buparvaquone

PONE-D-22-20200R2

Dear Dr. Karagenc,

We’re pleased to inform you that your manuscript has been judged scientifically suitable for publication and will be formally accepted for publication once it meets all outstanding technical requirements.

Kind regards,

Vikrant Sudan, PhD

Academic Editor

PLOS ONE

Additional Editor Comments (optional):

I am happy with the corrections incorporated and with reviewers comments. I recommend acceptance of manuscript.
---

## [Editor Report · Acceptance letter]

26 Dec 2022

PONE-D-22-20200R2 

Selection of genotypes harbouring mutations in the *cytochrome b* gene of *Theileria annulata* is associated with resistance to buparvaquone 

Dear Dr. Karagenc:

I'm pleased to inform you that your manuscript has been deemed suitable for publication in PLOS ONE. Congratulations! Your manuscript is now with our production department. 

Kind regards, 

on behalf of

Dr. Vikrant Sudan 

Academic Editor

PLOS ONE